# TopoCore: Unifying Topology Manifolds and Persistent Homology for Data Pruning

## Abstract

Geometric coreset selection methods, while practical for leveraging pretrained models, are fundamentally unstable. Their reliance on extrinsic geometric metrics makes them highly sensitive to variations in feature embeddings, leading to poor performance when transferring across different network architectures or when dealing with noisy features. We introduce TopoCore, a novel framework that resolves this challenge by leveraging the principles of topology to capture the intrinsic, stable structure of data. TopoCore operates in two stages, (1) utilizing a *topology-aware manifold approximation* to establish a global low-dimensional embedding of the dataset. Subsequently, (2) it employs *differentiable persistent homology* to perform a local topological optimization on the manifold embeddings, scoring samples based on their structural complexity. We show that at high pruning rates (e.g., 90%), our *dual-scale topological approach* yields a coreset selection method that boosts accuracy with up to $4\times$ better precision than existing methods. Furthermore, through the inherent stability properties of topology, TopoCore is (a) exceptionally robust to noise perturbations of the feature embeddings and (b) demonstrates superior architecture transferability, improving both accuracy and stability across diverse network architectures. This study demonstrates a promising avenue towards stable and principled topology-based frameworks for robust data-efficient learning.

## 1 Introduction

The computational demands of training modern deep learning systems have escalated dramatically due to the scale of contemporary models and datasets. This growth has made training and fine-tuning computationally prohibitive, creating a need for data-efficient learning strategies. Data pruning is one such strategy that aims to subsample a large dataset into a smaller, representative subset (or coreset) that preserves the essential learning characteristics of the full dataset. Thereby enabling rapid model training, efficient fine-tuning, and reduced storage costs, all while minimizing degradation in final model performance.

Broadly, coreset selection methods can be categorized into three major categories. *Optimization-based* methods select a coreset whose loss landscape (Killamsetty et al., 2021b; Mindermann et al., 2022) or gradi-

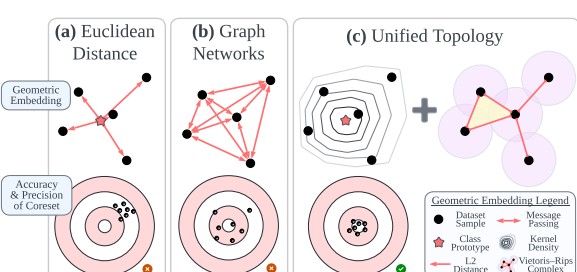

Figure 1: **Topological data selection yields higher-performing and more stable coresets.** A comparison of coreset performance across random seeds reveals the limitations of common methods. (a) Euclidean-based selection is stable but achieves lower accuracy. (b) Graph-based methods achieve higher accuracy but are highly variable. (c) Our topological approach achieves both high accuracy and stability.

ent dynamics (Mirzasoleiman et al., 2019; Killamsetty et al., 2021a; Tan et al., 2023) most closely align with that of the entire dataset, ensuring that a model trained on the subset exhibits comparable generalization. While often effective, such approaches are hampered by significant practical limitations such as their reliance on computationally intensive second-order (Pooladzandi et al.,

2022) or bilevel optimization (Borsos et al., 2020). *Score-based* methods rank and choose training samples based on model prediction scores. These can include scores derived from training dynamics (Toneva et al., 2019; Garg & Roy, 2023; Zheng et al., 2025) and uncertainty estimations (Paul et al., 2021; He et al., 2023; 2024; Cho et al., 2025). However, these scores are inherently model and training-dependent and reflect the knowledge of a specific network at a specific point in training. This makes optimization and score-based methods not only expensive, as they require training from scratch for some amount of time, but also incompatible with the vast and growing ecosystem of publicly available pretrained models, where only the final weights are accessible.

To overcome this constraint, *geometry-based* coreset selection methods can operate on static feature embeddings from a pretrained model. Approaches in this domain range from representing samples based on the penultimate-layer feature embedding space (Xia et al., 2023), measuring distributional similarity via optimal transport (Xiao et al., 2024) or Wasserstein distance (Xiong et al., 2024), or using the geometric reconstruction error of samples with decision boundary information (Yang et al., 2024). While these methods avoid costly training analysis, a significant limitation is their reliance on metrics that are sensitive to the extrinsic geometry of the feature embedding space, a vulnerability we term "geometric brittleness." This brittleness leads to two primary shortcomings: (1) a tendency to prioritize samples from dense regions at the expense of informative samples from the sparse tails of the distribution (Zheng et al., 2023), and (2) an instability in performance across different network architectures or when noise is introduced to the embeddings. This is most apparent in Euclidean-distance metrics (Xia et al., 2023) or message-passing graph methods (Maharana et al., 2024; Xie et al., 2025), which are highly sensitive to changes in the feature embedding space (see Fig. 1).

In this work, we introduce TopoCore, a novel framework that resolves the challenge of geometric brittleness by leveraging topological analysis (Seifert & Threlfall, 1980). At a high level, topology is a branch of algebraic geometry concerned with the properties of a space that are preserved under continuous deformations like stretching and bending, but not tearing. A classic example is that, in a topological sense, a coffee mug and a donut are equivalent as both possess a single hole. By focusing on this stable, intrinsic structure (the hole) rather than the transient, extrinsic geometric measurements (like distance or curvature), we can analyze the feature embeddings of datasets with a stable, topological metric. This allows TopoCore to achieve exceptional stability to slight perturbations in the feature embedding space, caused by noisy features or those arising from differences in feature embeddings across network architectures (Cohen-Steiner et al., 2005; Suresh et al., 2024). This enables the use of proxy models (Coleman et al., 2020) or the direct use of the vast corpus of pretrained and foundational models to generate coresets without the need for retraining a specific model from scratch.

Our framework first establishes a **global structure** by using *topology-aware manifold approximation* (McInnes et al., 2018a; Wang et al., 2021) to project high-dimensional features into standardized low-dimensional manifold embeddings. While this global structure can group similar samples, it fails to distinguish which samples to prioritize within a localized region. Existing methods often resort to random sampling within localized regions (Zheng et al., 2023) or use geometric heuristics like message-passing (Maharana et al., 2024). To better complement this global view with **local structure**, we then employ *differentiable persistent homology* (Scoccola et al., 2024; Carrière et al., 2024; Mukherjee et al., 2024) to assess a sample's structural relevancy, relative to its immediate neighbors. At a high level, persistent homology is a method that tracks the "birth" and "death" (persistence) of topological structures at multiple scales (filtration of a homology group). For our application, we perform an optimization that *maximizes the persistence (the birth and death time) of local topological features* constructed from the filtration of the manifold projected Vietoris–Rips complex (Loiseaux et al., 2023a). This process iteratively repositions samples to an optimal configuration that resolves topological ambiguities and enhances topological stability, directly measuring a sample's contribution to the structural complexity of its local neighborhood. Finally, a unified score consisting of both the global density (of the manifold embeddings) and the local persistence are combined to provide a balance between global and local topological structures.

Our approach makes the following contributions:

- We introduce TopoCore, a novel coreset selection framework that defines sample importance through a dual-scale topological analysis. It combines a *global manifold projection* with a *local persistence* score derived from a differentiable persistent homology optimization to

identify structurally critical samples with higher accuracy and stability compared to previous coreset methods.

- We demonstrate that TopoCore is more stable across diverse network architectures (e.g., from ResNet to ViT), enabling the use of small proxy models or off-the-shelf pretrained models to generate high-quality coresets without costly retraining.

- We provide extensive empirical validation showing that TopoCore significantly outperforms state-of-the-art geometric, gradient, and score-based methods. Our framework delivers coresets with higher accuracy and precision (up to $4\times$ better), and is substantially more robust to noisy feature embeddings (up to $8\times$ better precision), especially at high data pruning rates.

By defining sample importance through the stable, intrinsic properties of topology, TopoCore moves beyond brittle geometric metrics to deliver a truly robust coreset framework.

## 2 BACKGROUND AND RELATED WORK

### 2.1 TOPOLOGY AT TWO SCALES

While many modern topological algorithms inherently model both the global and local structure of data simultaneously, our work decouples these concepts into two distinct stages. For the purposes of this paper, we define global topology as the overall manifold structure of the entire dataset, which we capture as a low-dimensional embedding. We then define local topology as the fine-grained structure arising from the interactions between a single sample and its immediate neighbors, which we analyze using persistent homology.

#### 2.1.1 LOW-DIMENSIONAL MANIFOLD APPROXIMATIONS

A critical step in high-dimensional analysis is creating a low-dimensional data representation. Linear methods like Principal Component Analysis (PCA) (Pearson, 1901) are efficient but preserve only global variance, failing to capture complex non-linear structures. In contrast, non-linear techniques like t-Distributed Stochastic Neighbor Embedding (t-SNE) (van der Maaten & Hinton, 2008) excel at preserving fine-grained local neighborhoods, but this focus often distorts the data's overall global structure. More recent techniques directly leverage principles from topology to create more faithful manifold embeddings. State-of-the-art methods like Uniform Manifold Approximation and Projection (UMAP) (McInnes et al., 2018a) and PaCMAP (Wang et al., 2021) model high-dimensional data as a fuzzy topological structure to preserve both fine-grained local connectivity and large-scale global relationships. As shown by de Bodt et al. (2025), these topology-aware techniques produce low-dimensional manifold embeddings with more compact and well-defined clusters, which enhances the impact of downstream analysis.

These methods have proven highly effective for interpreting the complex representations learned by deep models across numerous domains, from single-cell genomics (Becht et al., 2018) to clustering in dictionary learning (Fel et al., 2023; 2024). While the quantitative fidelity of such embeddings is an area of active research (Jeon et al., 2025), we believe that they are well-suited for our task. By explicitly preserving the nearest-neighbor structure from the high-dimensional space, methods such as UMAP and PacMAP inherently maintain the density landscape of the data manifold. This directly preserves the notions of "prototypicality" (samples in high-density regions) and "atypicality" (samples in sparse regions), making the embedding a reliable foundation for our subsequent sample importance scoring (see Section A.1 for more detail).

#### 2.1.2 INTERACTIONS OF SAMPLES AND THEIR NEAREST NEIGHBORS

Understanding the local interactions between a sample and its neighbors is crucial for determining its importance. A prevalent approach is the use of Graph Neural Networks (GNNs), which propagate information between nodes on a graph typically defined by nearest-neighbor relationships. In GNNs, a sample's importance is quantified through a learned message-passing function that aggregates features from its local neighborhood (Maharana et al., 2024). Other recent methods use graph-level structural entropy combined with Shapley values and blue noise sampling (Chen et al., 2014) to

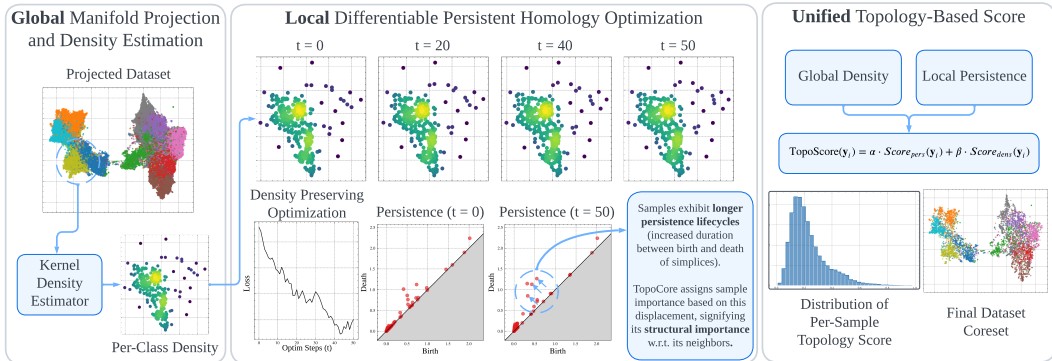

Figure 2: **An overview of TopoCore**. **(Left)** A topology-aware projection visualizes the *global* data manifold. **(Middle)** Within each class, a density-preserving persistent homology optimization derives a *local* persistence score per sample. The color map indicates high (yellow) to low (blue) density. **(Right)** The final coreset is constructed via stratified sampling on a unified score combining *global density* and *local persistence*. This not only prioritizes the most topologically informative samples but also faithfully represents the density distribution of the original dataset.

select a diverse coreset (Xie et al., 2025). However, these approaches operate on a single, fixed graph structure and can be sensitive to the geometric hyperparameters used in its construction.

In contrast, persistent homology (Botnan & Lesnick, 2022) offers a fundamentally different and more robust framework. Instead of analyzing a single graph, it studies the evolution of higher-order topological structures (e.g., connected components, loops, voids) across a multi-scale filtration of simplicial complexes. This provides a complete summary of the data's shape at all scales simultaneously. A key advantage of persistent homology is its proven stability (Cohen-Steiner et al., 2005). The persistence diagram of a dataset is guaranteed to change only slightly in response to small perturbations of the input data, making it a robust descriptor of local structure (Turkes et al., 2022; Mishra & Motta, 2023). Inclusion of these robust geometric descriptors has been widely used for understanding feature embeddings in machine learning applications, including but not limited to topologically stable autoencoders (Moor et al., 2020), monitoring generalization in networks over training (Birdal et al., 2021), and exploring topological changes of data throughout a network's layers (Naitzat et al., 2020). For a detailed overview on the underlying construction of simplicial complexes and persistent homology, please see Section A.2.

While traditional persistent homology provides a powerful descriptive tool, its integration into modern deep learning pipelines has been limited as it is not inherently differentiable. Recent advances in differentiable persistent homology have overcome this barrier by enabling the backpropagation of gradients from the persistence diagram back to the coordinates of the input data points (Carrière et al., 2024; Mukherjee et al., 2024). This allows for the direct optimization of the data's topological features within a gradient-based framework. The work of Scoccola et al. (2024) provides a fast and stable computational framework for these gradients, even for the more expressive case of multiparameter persistent homology. By leveraging this, we can move beyond simply describing the local topology and instead perform an optimization to actively enhance it. This allows us to define a sample's local importance by directly measuring its role in the topological stability of its local neighborhood.

## 3 METHODOLOGY

Our proposed method, TopoCore, constructs a coreset by analyzing the data's topological structure at two distinct scales. (1) *Global Manifold Embedding*, projects the original high-dimensional embeddings into a standardized low-dimensional space. This ensures a stable, global view of the data's overall structure. (2) *Local Topological Interaction*, which employs differentiable multiparameter persistent homology to probe the local structure formed by samples and their closest neighbors. Together, these two topological scales are used to derive an importance score for each sample based on global density and local persistent homology, contributing to a unified topological measurement for selecting individual samples (see Fig. 2).

## 3.1 GLOBAL STRUCTURE: DATASET REPRESENTATION WITH TOPOLOGICAL MANIFOLD EMBEDDING

Given a well-trained deep model, denoted by $f(\cdot)$, we can express it as a composition of a feature extractor $h(\cdot)$ and a classifier $g(\cdot)$, such that $f(\cdot) = g(h(\cdot))$. Here, $h(\cdot)$ represents the network up to the *penultimate layer*, which maps an input data point $\mathbf{x}$ to a high-dimensional feature embedding $\mathbf{z} = h(\mathbf{x}) \in \mathbb{R}^D$. The full training dataset $S = \{(\mathbf{x}_1, y_1), \ldots, (\mathbf{x}_N, y_N)\}$ can thus be transformed into a high-dimensional feature set $Z = \{\mathbf{z}_1, \ldots, \mathbf{z}_N\}$. While this high-dimensional space $Z$ contains rich semantic information, its extrinsic geometry is often complex and architecture-dependent. To obtain a stable and standardized representation, we project $Z$ onto a low-dimensional manifold using topology-based manifold approximation and projection techniques (McInnes et al., 2018a; Wang et al., 2021).

This process involves two main stages. First, a topological representation of the high-dimensional data is constructed as a fuzzy simplicial set. This structure captures the data's shape by assigning a membership strength $(p_{ij})$, to the potential connections between each point and its neighbors, where the "fuzzy" aspect represents the belief that a certain simplex exists in the true underlying manifold. Subsequently, a low-dimensional embedding is learned $Y = \{\mathbf{y}_1, \ldots, \mathbf{y}_N\}$, where $\mathbf{y}_i \in \mathbb{R}^d$ and $d \ll D$, whose own fuzzy simplicial set $(q_{ij})$ is similarly defined. The final low-dimensional representation $Y$ is found by optimizing the positions of the points $\{\mathbf{y}_i\}$ to minimize a cross-entropy loss between the high-dimensional $(p_{ij})$ and low-dimensional $(q_{ij})$ pairwise similarities:

$$\mathcal{L}_{\text{proj}}(Y) = \sum_{i \neq j} \left[ p_{ij} \log \left( \frac{p_{ij}}{q_{ij}} \right) + (1 - p_{ij}) \log \left( \frac{1 - p_{ij}}{1 - q_{ij}} \right) \right] \tag{1}$$

This process yields a standardized manifold embedding that preserves the data's intrinsic shape. Through a detailed investigation into different manifold approximation and projection techniques presented in Section A.3 we use the UMAP (McInnes et al., 2018a) algorithm as it creates more uniform manifold embeddings across network architectures. On this low-dimensional manifold, we compute a **Density Score** for each sample using a Kernel Density Estimator (KDE), to capture its global representativeness. This estimates the probability density at each sample $\mathbf{y}_i$ by summing the contributions of neighboring points $\sum_{j=1}^{n} \mathbf{y}_j$:

$$\text{Score}_{\text{dens}}(\mathbf{y}_i) = \frac{1}{Nh} \sum_{j=1}^{N} K \left( \frac{\mathbf{y}_i - \mathbf{y}_j}{h} \right) \tag{2}$$

where $N$ is the number of samples, $K$ is a Gaussian kernel, and $h$ is the bandwidth parameter. This score allows us to distinguish samples in high-density (prototypical) regions from those in low-density (atypical) regions of the manifold.

## 3.2 LOCAL STRUCTURE: SAMPLE NEIGHBORHOODS WITH PERSISTENCE-BASED OPTIMIZER

The *Global Manifold Embedding* provides a low-dimensional representation that faithfully preserves the global structure of the data manifold. While this ensures a stable, high-level representation, a purely global perspective is insufficient for identifying the most informative samples, whose importance is often defined by the complex local interactions with their nearest neighbors. To capture this fine-grained structure, we leverage persistent homology not as a static descriptor, but as the foundation for a dynamic topological optimization process. The objective of this process is to iteratively adjust the position of each point within its class manifold to maximize topological persistence. This is performed independently for each class $c \in \{1, \ldots, C\}$ to analyze the specific intra-class structure. For each class, we begin with its point cloud from the global manifold embedding $Y_c = \{\mathbf{y}_i \mid \text{label}(\mathbf{y}_i) = c\}$ and construct a Vietoris-Rips filtration (Oudot, 2015) on $Y_c$ due to it's computational scalability compared to Alpha and Čech complexes, as shown in Otter et al. (2017) and Mishra & Motta (2023).

Similar to work from Scoccola et al. (2024) we define a differentiable loss function, $\mathcal{L}_{\text{pers}}(Y_c)$, whose negative gradient, $-\nabla_{Y_c} \mathcal{L}_{\text{pers}}$, points in the direction that *maximally increases the total persistence of the features*. This loss is formulated using a multi-parameter filtration considering two parameters: (1) the class-manifold Vietoris-Rips filtration $(VR_{Y_c})$ and (2) the class-manifold Kernel Density

Estimator ($\hat{f} = KDE_{Y_c}$). The persistence of this two-parameter filtration is summarized using the Hilbert decomposition signed measure of the $H_1$ persistence module, denoted $\mu^{Hil}_{H_1(VR_{Y_c},\hat{f})}$ (Loiseaux et al., 2023b). In simple terms, this descriptor represents the persistence diagram as a finite collection of positive point masses (representing feature births) and negative point masses (representing feature deaths) in the parameter space of (distance, density). Our objective is to maximize the persistence, which is accomplished by maximizing the Optimal Transport (OT) distance between this signed measure and the zero measure, $\mathbf{0}$ (Carriere et al., 2021). The differentiable loss function for a given class $c$ is therefore defined as:

$$\mathcal{L}_{\text{pers}}(Y_c) := \text{OT}(\mu^{Hil}_{H(VR_{Y_c},\hat{f})}, \mathbf{0}) \tag{3}$$

The optimization seeks a new point configuration $Y_c'$ that minimizes this loss, solved iteratively via gradient descent (see Section A.4 exploring the number of optimization steps). This specific formulation ensures that the optimization *enhances topological stability while preserving the original density of the class manifold*, as the density is recomputed at each epoch and is an integral part of the loss calculation. We then define the **Persistence Score** for each sample $\mathbf{y}_i$ belonging to class $c$ as the magnitude of its total displacement during its class-specific optimization, where $\mathbf{y}_i$ is the initial position and $\mathbf{y}'_i$ is the final, optimized position.:

$$\text{Score}_{\text{pers}}(\mathbf{y}_i) = \|\mathbf{y}_i - \mathbf{y}'_i\|_2, \quad \text{for } \mathbf{y}_i \in Y_c, \mathbf{y}'_i \in Y_c' \tag{4}$$

**Interpreting this notion of local dataset structure.** A high Persistence Score quantifies the degree of topological instability a sample introduces within its own class manifold. Crucially, our optimization process is designed to be density-preserving, it enhances local topological features without altering the overall density distribution of the class manifold. This is vital for coreset selection, as it ensures our search for structurally important samples does not distort the global representativeness of the data. The optimization process repositions these points to clarify the underlying intra-class structure and increase its persistence. Therefore, the magnitude of this corrective displacement serves as a direct, dynamic measure of a sample's contribution to the topological complexity of its class, derived from the collective interaction of every point in the manifold.

### 3.3 COMPREHENSIVE SCORE WITH GLOBAL AND LOCAL DATASET STRUCTURES

To create a comprehensive sample importance metric, we formulate a final score that unifies the global density and local persistence information as a weighted combination of these two metrics:

$$\text{TopologyScore}(\mathbf{y}_i) = \alpha \cdot \text{Score}_{\text{pers}}(\mathbf{y}_i) + \beta \cdot \text{Score}_{\text{dens}}(\mathbf{y}_i) \tag{5}$$

where hyperparameters $\alpha, \beta \in [0, 1]$ modulate the influence of local topological complexity (Persistence Score) versus global distributional rarity (Density Score). From an exploration across different ranges of $\alpha$ and $\beta$ in Section A.5 we find that $\alpha = 0.5$ and $\beta = 0.5$ provide a good balance between global and local information. This allows our framework to construct a coreset that is not only rich in challenging, boundary-defining examples but also maintains a faithful representation of the full dataset's underlying distribution. For a detailed illustrative example please see Section A.6.

### 3.4 A TRAINING-FREE PROXY FOR MISLABELED SAMPLES

Inspired by findings in CCS (Zheng et al., 2023), we ensure our coreset is not corrupted by noisy or mislabeled data, which can receive high importance scores yet degrade model performance (Swayamdipta et al., 2020) by incorporating a final filtering step. While most recent methods, including CCS and D2 (Maharana et al., 2024), use metrics like Area Under the Margin (AUM) (Pleiss et al., 2020) to identify mislabeled samples by tracking their scores throughout training, this approach diverges from the ultimate objective to develop methods that operate solely on static, pretrained models.

To overcome this, we propose a training-free proxy for AUM, which we term the Neighborhood Label Purity Score (NLPS). For each sample, we compute the fraction of its k-nearest neighbors in the feature space that share its class label. A low NLPS indicates that a sample resides in a mixed-label neighborhood, suggesting it is on a noisy decision boundary or potentially mislabeled, analogous to the "flip-flop" candidates identified by AUM. We validate this approach by testing out multiple decision boundary flip-flop proxies, as shown in Section A.7.

Table 1: Comparison of the test accuracy across various coreset selection methods on CIFAR-10, CIFAR-100 and ImageNet-1K. TopoCore (w/ NLPS) consistently outperforms other training-free methods. While *TopoCore's* topological advantage is most pronounced on challenging datasets like ImageNet-1K and at high pruning rates (e.g., 90%). For standard deviation values please see Table 5 in Section A.9.

| | | CIFAR-10 (ResNet-18) | | | | | CIFAR-100 (ResNet-18) | | | | | ImageNet-1K (ResNet-50) | | | | |
|---|---|---|---|---|---|---|---|---|---|---|---|---|---|---|---|---|
| | Pruning Rate ($\rightarrow$) | 30% | 50% | 70% | 80% | 90% | 30% | 50% | 70% | 80% | 90% | 30% | 50% | 70% | 80% | 90% |
| No Training Dynamics | Random | 94.5 | 93.5 | 90.8 | 86.6 | 76.7 | 75.3 | 71.6 | 63.7 | 55.9 | 34.0 | 69.8 | 68.4 | 65.1 | 61.9 | 52.5 |
| | Moderate (Xia et al., 2023) | 94.2 | 93.1 | 89.9 | 87.2 | 76.9 | 74.9 | 70.1 | 63.7 | 56.1 | 34.9 | 69.5 | 65.8 | 60.5 | 57.7 | 50.0 |
| | FDMat (Xiao et al., 2024) | 94.7 | 93.6 | 90.8 | 87.3 | 74.4 | 75.4 | 71.9 | 64.0 | 56.1 | 37.5 | 70.8 | 68.7 | 65.5 | 62.0 | 51.9 |
| | **TopoCore (w/ NLPS)** | 94.8 | 93.6 | 90.3 | 87.3 | 77.1 | 75.6 | 71.9 | 65.3 | 56.7 | 41.6 | 70.7 | 69.9 | 66.4 | 63.2 | 53.9 |
| With Training Dynamics | Moderate (w/ AUM) | 93.9 | 93.1 | 90.1 | 87.1 | 79.9 | 75.9 | 72.4 | 66.7 | 60.2 | 40.0 | 69.6 | 67.2 | 63.9 | 60.4 | 52.7 |
| | Forgetting (Toneva et al., 2019) | 94.5 | 92.6 | 89.8 | 85.6 | 67.6 | 74.8 | 67.2 | 50.6 | 32.3 | 24.3 | 69.9 | 66.8 | 60.2 | 59.1 | 50.0 |
| | Glister (Killamsetty et al., 2021b) | 94.4 | 93.8 | 90.8 | 85.1 | 66.8 | 75.8 | 70.7 | 66.1 | 54.7 | 38.4 | 66.3 | 63.5 | 59.3 | 56.5 | 49.3 |
| | LCMat-S (Shin et al., 2023) | 94.5 | 93.3 | 90.5 | 86.9 | 75.1 | 75.3 | 71.1 | 62.5 | 55.1 | 36.1 | 69.8 | 67.5 | 62.2 | 59.7 | 48.8 |
| | CCS (Zheng et al., 2023) | 95.5 | 94.8 | 93.0 | 90.7 | 81.9 | 76.9 | 73.8 | 67.8 | 60.7 | 45.2 | 70.1 | 69.1 | 65.7 | 62.6 | 55.2 |
| | D2 (Maharana et al., 2024) | 95.6 | 94.8 | 93.1 | 89.2 | 80.9 | 75.1 | 71.2 | 67.8 | 61.1 | 44.3 | 69.5 | 67.1 | 65.7 | 62.7 | 55.5 |
| | **TopoCore** | 94.7 | 93.7 | 91.6 | 88.7 | 82.1 | 75.9 | 72.8 | 66.9 | 61.9 | 45.8 | 70.8 | 69.5 | 66.2 | 63.1 | 56.1 |

## 3.5 Topological Stratification and Coreset Construction

In the final phase, we construct the coreset from the remaining "clean" dataset (post-AUM or NLPS filtering). We perform stratified sampling based on the calculated TopologyScore to select samples while strictly preserving the original class distribution for a desired pruning rate. Importantly, TopoCore generates an *unbalanced* coreset, rather than enforcing a uniform number of samples per class, we respect the intrinsic class imbalance of the original dataset structure. This process yields a final subset that is topologically rich, globally representative, and robust to label noise.

In summary, the complete TopoCore workflow proceeds through three distinct phases: (1) *Dual-scale topological scoring* (projecting global manifolds and optimizing local persistence), (2) *Mislabeled sample filtering* (creating a "clean" dataset removing mislabeled samples via AUM or NLPS), and (3) *Topological selection* (stratified sampling on the "clean" dataset based on the unified topology score).

## 4 Results

### 4.1 Experimental Setup

TopoCore utilizes several tools and frameworks. Manifold projection is performed using `UMAP` (McInnes et al., 2018b), `multipers` (Loiseaux & Schreiber, 2024) facilitates differential persistent homology which uses the `Gudhi` C++ library (Maria et al., 2025) as a backend, and `DeepCore` (Guo et al., 2022) is used to standardize coreset selection and training across different methods.

To ensure fair comparisons in our experiments, we evaluate two versions of our framework. When benchmarking against other static methods, we use *TopoCore (w/ NLPS)*, which incorporates training-free NLPS for mislabeled samples. When comparing against methods that require training-time information, we use *TopoCore*, which incorporates the original AUM score. We compare **TopoCore (w/NLPS)** with several static geometry-based coreset selection methods: A) **Random** selection. B) **Moderate** (Xia et al., 2023) uses samples near the median distance to a class prototype (the barycenter of a point-mass distribution). C) **FDMat** (Xiao et al., 2024) matches data distribution between dataset and coreset using optimal transport. We compare **TopoCore** with several geometry, score, and optimization-based methods that require training-time information: D) **Moderate + AUM** incorporating Moderate with mislabeled sample removal with AUM. E) **Forgetting** (Toneva et al., 2019) uses number of times an example is incorrectly classified after being correctly classified earlier during training. F) **Glister** (Killamsetty et al., 2021b) uses bi-level optimization. G) **LCMat-S** (Shin et al., 2023) matches loss curvature between dataset and coreset. H) **CCS** (Zheng et al., 2023) uses stratified sampling of difficulty scores (such as AUM or Forgetting) with intra-strata random sampling. H) **D2** (Maharana et al., 2024) uses a message-passing graph network while also incorporating AUM for mislabeled samples. All reported coreset accuracies and standard deviations (except for ImageNet-1k) are computed over five independent training runs, with each run using a different random seed.

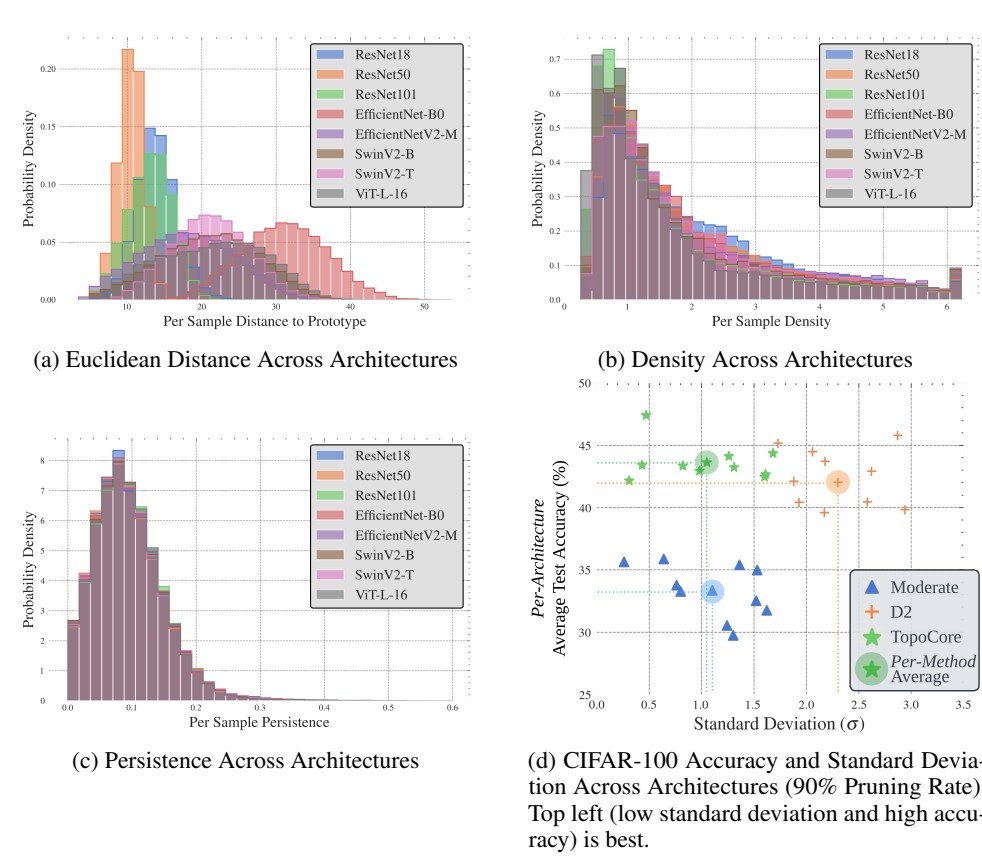

(a) Euclidean Distance Across Architectures

(b) Density Across Architectures

(c) Persistence Across Architectures

(d) CIFAR-100 Accuracy and Standard Deviation Across Architectures (90% Pruning Rate). Top left (low standard deviation and high accuracy) is best.

Figure 3: Topological metrics provide better stability across network architectures, which translates directly to better coreset performance. Metric distributions become progressively more uniform as we move from (a) unstable Euclidean distances, to (b) density estimation from global topological projection, and finally to (d) local persistence. This enhanced metric stability allows TopoCore to consistently outperform geometry-based baselines (d), achieving both higher mean accuracy and lower standard deviation across 10 diverse architectures at a high pruning rate of 90%.

## 4.2 Higher Performing and more Stable Coresets with TopoCore

Our results in Table 1 show that TopoCore consistently generates high performing coresets compared to existing methods across CIFAR-10, CIFAR-100, and ImageNet-1K. Across the majority of dataset and pruning rate scenarios, TopoCore (w/ NLPS) outperforms all other training-free geometric methods. While TopoCore's performance is competitive on simpler datasets and at lower pruning rates, its advantage becomes most pronounced on more challenging datasets like ImageNet-1K and at the highest pruning rates, where it is consistently the top-performing method. Beyond mean accuracy, TopoCore (w/ NLPS) and TopoCore also demonstrate superior stability. As shown by the standard deviation values in Table 5 in the appendix (Section A.9), our method produces coresets with a significantly higher precision (up to 4×) across different runs. This high degree of stability is a critical practical advantage, as it ensures reliable, high-quality performance without requiring users to run multiple costly trials to mitigate the effects of randomness.

## 4.3 Stability to Changing Feature Embeddings across different Networks

We investigate the stability of different geometric metrics across a wide range of network architectures, finding that stability and transferability increases dramatically as we move from geometric to topology-based metrics. First, we analyze the Euclidean distance of each sample to its class prototype, which is highly inconsistent across architectures, demonstrating the "geometric brittleness" of this metric (Fig. 3a). Next, by examining our Density Score, global topological structure via the kernel density of our manifold embedding, we observe a significant improvement in the metric uniformity (Fig. 3b).

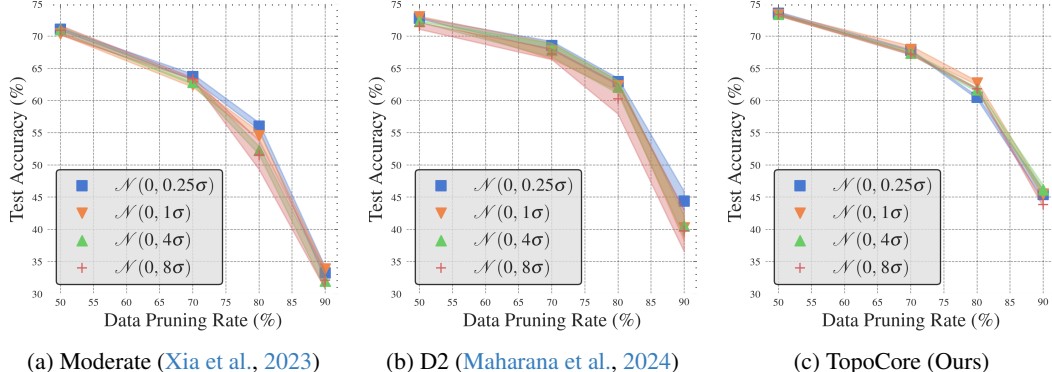

Figure 4: Comparison of Moderate, D2, and TopoCore when adding certain levels of noise to the feature embeddings on CIFAR-100. (a) Moderate has decent precision (shown by the tightness of the shaded region) but lower accuracy on average. (b) D2 has higher accuracy but significantly looser precision, especially at high pruning rate and high $\epsilon$-noise. (c) TopoCore achieves both high accuracy and tighter standard deviation across all pruning rates and $\epsilon$-noise.

Finally, by analyzing the local topological structure with our Persistence Score, we achieve a remarkable level of stability where this metric almost completely aligns across all tested architectures (Fig. 3c). This superior metric stability translates directly to coreset performance (Fig. 3d), TopoCore consistently yields higher-performing and more stable coresets (higher accuracy, lower standard deviation) compared to other geometric methods, with the most significant advantage at high data pruning rates. Please see Table 6 for exact values and for results on different coreset pruning rates and Section A.8 for further justification.

**Implications.** The superior architectural transferability of our method has practical implications beyond the use of small proxy models Coleman et al. (2020). It is a step towards the creation of reusable, model-agnostic coresets, where a dataset can be curated once with an efficient model and then used to train or benchmark a multitude of diverse, larger architectures. This approach aligns with recent work that uses topological measures, such as Betti numbers, to investigate and compare the complexity of embedding spaces across different networks (Suresh et al., 2024).

## 4.4 STABILITY TO NOISY FEATURE EMBEDDINGS

To evaluate the robustness of our method to perturbations in the feature space, we inject Gaussian noise into the penultimate layer embeddings of a ResNet-18 model on CIFAR-100. The magnitude of the noise for each sample is scaled by the standard deviation of its own feature components. We compare TopoCore against geometry-based baselines, Moderate Xia et al. (2023) and D2 Maharana et al. (2024), which are sensitive to such perturbations. "Noisy" versions of the feature embeddings are created by perturbing each sample's feature vector, $\mathbf{z} \in \mathbb{R}^D$. This is done by first computing the per-sample standard deviation of its feature components, $\sigma_{\mathbf{z}}$, then adding a noise vector $\epsilon \sim \mathcal{N}(0, \sigma_{\mathbf{z}})$ to the original features $\mathbf{z}' = \mathbf{z} + \epsilon$.

As shown in Fig. 4, TopoCore demonstrates significantly higher resilience to noise across all tested levels. While the performance of the baseline methods degrades with increased perturbation, our topological approach maintains high accuracy and stability. This robustness is particularly evident at high pruning rates, highlighting our method's ability to identify a stable coreset even in a noisy feature space. For detailed quantitative results, see Table 3 in the appendix.

**Implications.** This robustness to noisy features indicates our method is not overly dependent on a perfectly optimized source model. This suggests that effective coresets could be generated using embeddings from models that are partially-trained, quantized for edge devices, or applied to slightly out-of-distribution data as similarly shown in Turkes et al. (2022). This flexibility broadens the potential applicability of our approach to a wider range of practical scenarios.

## 5 CONCLUSION

In this work, we addressed the critical challenge of instability in geometric coreset selection methods, which arises from their reliance on extrinsic metrics. We presented TopoCore, a novel framework that overcomes this "geometric brittleness" by leveraging topology to capture the data's intrinsic structure. Our dual-scale topological approach combines a global topology-aware manifold projection with a local importance score derived from differentiable persistent homology. TopoCore exhibits several key advantages: (1) yields coresets with higher accuracy and stability, (2) is more stable across many different network architectures, and (3) is highly robust to noisy feature embeddings.

The stability of TopoCore makes it useful in practical applications. For the training of large foundational models, it curtails the "random seed lottery," de-risking the effects of instability by ensuring a consistently high-quality data subset. In automated pipelines, its more deterministic nature provides the auditability and reliability essential for building trustworthy, maintainable and safety-critical AI systems. By delivering both accuracy and stability, TopoCore marks a step towards feasible and principled topology-based framework's for data-efficient learning.

## REPRODUCIBILITY STATEMENT

We strongly believe in the importance of reproducibility in scientific research. Therefore, we strive for full transparency in our work. To this end regarding explanations in the paper, the Methodology section (Section 3) provides a comprehensive overview of our dual-scale topology-based method. The Experimental Setup subsection (Section 4.1) offers further details regarding libraries and tools needed to facilitate reproducibility. All quantitative values from experiments are provided in Section A.9. All manifold projection, persistent homology and network training hyperparameters as well as dataset mislabel ratios are also documented in Section A.10. To further promote reproducibility, we plan to publicly release our code on GitHub prior to the conference date, enabling readers to verify our results and utilize these methods in their own research. During the review process, an anonymous GitHub repository will be made available to reviewers and area chairs at the time of discussion, as per the ICLR Author Guidelines.

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

# A    APPENDIX

TABLE OF CONTENTS

## A.1    RELATING TOPOLOGY-BASED MANIFOLD EMBEDDING AND SAMPLE PROTOTYPICALITY

We qualitatively examine the link between the intra-class density of our projected manifold and the established notion of sample memorization (Feldman, 2020; Feldman & Zhang, 2020), measured via the input curvature score (Garg et al., 2024). Our findings reveal a clear correspondence: *high-density*, prototypical samples consistently exhibit *low input curvature* (characteristic of un-memorized examples), while *low-density*, atypical samples show high *input curvature* (a key indicator of memorization). This alignment demonstrates that our topological manifold projection effectively preserves the global structure that distinguishes prototypical from atypical samples.

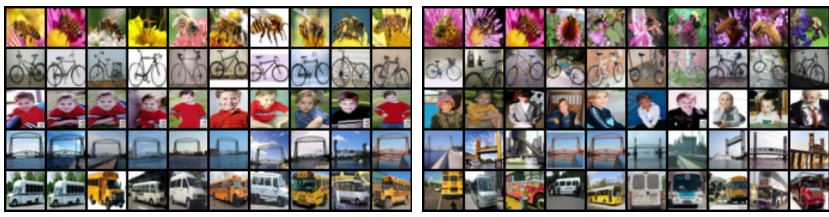

Figure 5: **Prototypical Samples:** Top-10 lowest curvature samples (left) vs. highest density samples (right) of the same class, for five CIFAR-100 classes.

Figure 6: **Atypical Samples:** Top-10 highest curvature samples (left) vs. lowest density samples (right) of the same class, for five CIFAR-100 classes.

## A.2 OVERVIEW OF SIMPLICIAL COMPLEXES AND PERSISTENT HOMOLOGY

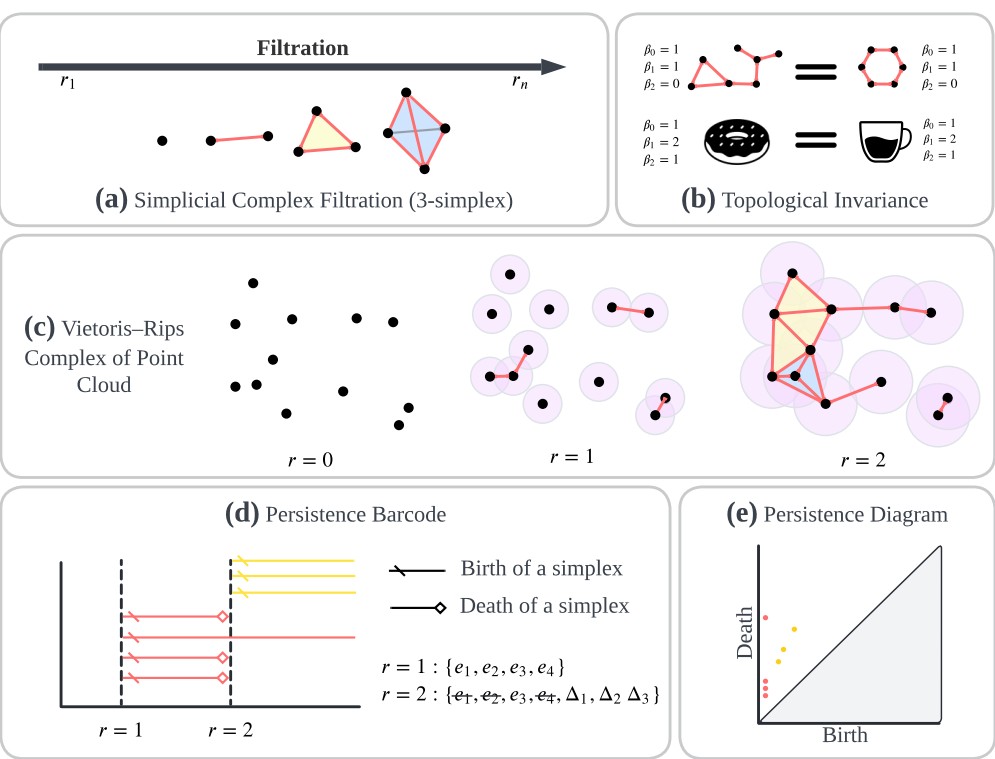

Figure 7: Overview of Simplicial Complexes and Persistent Homology

Persistent homology is used to characterize the topological variations in the shape of a finite metric space across multiple scales. At a high level, this can be described as the "birth" and "death" (persistence) of topological structures (defined by a homology group). The process begins by constructing a *simplicial complex*, a collection of points (0-simplices), edges (1-simplices), triangles (2-simplices), and their higher-dimensional counterparts that represents the data's structure (Boissonnat & Maria, 2014). To analyze how this structure changes with scale, a *filtration* is created (see Fig. 7(a)). This is a nested sequence of simplicial complexes, $K_{r_1} \subseteq K_{r_2} \subseteq \cdots \subseteq K_{r_n}$, indexed by a non-decreasing scale parameter $r$. For each complex $K_r$ in the filtration, we can compute its *homology groups*, $H_k(K_r)$, which are vector spaces that algebraically capture its $k$-dimensional features.

The rank of this group, known as the $k$-th *Betti number* ($\beta_k = \text{rank}(H_k(K_r))$), provides a count of these features: $\beta_0$ counts connected components, $\beta_1$ counts loops or tunnels, $\beta_2$ counts voids, and so on. These values are central to understanding the distinction between an object's *extrinsic* geometry and its *intrinsic* topological properties. A classic example which illustrates this difference is that of a coffee mug and a torus (donut). These two objects are topologically equivalent because they share the same Betti numbers (see Fig. 7(b)). Although their extrinsic geometries (including their shape,

curvatures, and distances as embedded in 3D space are very different), their intrinsic topology is identical. This is because one can be continuously deformed into the other without tearing or gluing, preserving the single hole that defines them both.

Persistent homology is not wedded to any form of metric construction, and in fact you can do persistence on purely abstract simplicial complexes and any filtration on it. However, for the purposes of our application, we apply this to a point cloud $P = \{\mathbf{x}_i\}$ using the common method of building a filtration with the *Vietoris-Rips (VR) complex* (see Fig. 7(c)). For a given scale $r \geq 0$, the complex $VR(P, r)$ contains all simplices $\sigma \subseteq P$ such that the Euclidean distance between any two points in $\sigma$ is at most $2r$. As $r$ increases, simplices are added to the complex, causing new components to merge with older components. **Persistent homology tracks the birth and death of these topological features throughout the filtration.** The inclusion map $K_{r_i} \hookrightarrow K_{r_j}$ for $r_i \leq r_j$ induces a homomorphism between the homology groups, $H_k(K_{r_i}) \to H_k(K_{r_j})$. A feature is said to be "born" at a scale $r_{\text{birth}}$ when it first appears and "dies" at a scale $r_{\text{death}}$ when it merges with an older feature visualized by the persistence barcode (Fig. 7(d)).

**Definition A.1** (Vietoris-Rips Filtration). *For a point cloud $P \subset \mathbb{R}^n$ and a scale parameter $r \geq 0$, the **Vietoris-Rips complex** $VR(P, r)$ is the simplicial complex whose vertices are the points in $P$ and whose simplices are all finite subsets of $P$ with a diameter of at most $2r$. A filtration is the nested sequence of complexes $\{VR(P, r)\}_{r \geq 0}$.*

The output of this process is summarized in a *persistence diagram* $\text{Dgm}(P)$, a multiset of points in the plane where each point corresponds to a single topological feature plotted at its (birth, death) $\to (b, d)$ coordinates (see Fig. 7(e)). The *persistence* of a feature is defined as its lifespan, $d - b$. **Points in the diagram that are further from the diagonal line $y = x$ represent robust, structurally significant features of the data**, while points close to the diagonal are interpreted as topological noise with short lifespans. This provides a stable, multi-scale signature of the data's underlying shape.

**Definition A.2** (Persistence Diagram). *Applying the homology functor $H_k(\cdot)$ (for a fixed dimension $k$, e.g., $k = 0$ for connected components) to a filtration yields a set of birth-death pairs $(b, d)$ representing the scales at which topological features appear and disappear. This multiset of pairs is the **persistence diagram**, denoted $\text{Dgm}(P)$. The **persistence** of a feature $(b, d)$ is defined as $d - b$.*

Please note that for clarity and ease of visualization in this overview, we present the 1-parameter persistence analysis. It is important to note, however, that our method, TopoCore, employs a more general multi-parameter persistence module, which is more complex to visualize but provides a richer description of the data's topology.

## A.3 TOPOLOGY-BASED MANIFOLD PROJECTION AS A METRIC STANDARDIZATION

Applying a UMAP projection as a preprocessing step is critical for achieving metric stability across diverse neural network architectures. While high-dimensional embeddings may vary in their extrinsic geometry, they share a common intrinsic topology. UMAP leverages this shared structure to construct a new, low-dimensional manifold that is not only topologically faithful but also geometrically standardized. A key consequence of this process is that the resulting low-dimensional embeddings are *density-preserving* across architectures. This standardization ensures that the global density score, a core component of our sample importance calculation, is a stable and reliable metric regardless of the source network.

To further elaborate the standardization of topology-based manifold approximation and projection across perturbations in the embedding space we look at correlation (Fig. 8) and distributions (Fig. 9) of per-sample distance to prototypes across different manifold projection and feature reduction techniques (a) PCA (Pearson, 1901), (b) t-SNE (van der Maaten & Hinton, 2008) (c) PaCMAP (Wang et al., 2021) and (d) UMAP (McInnes et al., 2018a). We see that the topology-based methods, UMAP and PaCMAP, demonstrate significantly higher correlation and thus better transferability across architectures compared to linear PCA or the more locally-focused t-SNE. Notably, UMAP exhibits slightly superior transferability over PaCMAP, reinforcing its selection for our framework. This high correlation between smaller models (e.g., ResNet-18) and larger models is particularly valuable, as it validates the use of computationally inexpensive networks to generate manifold embeddings that remain effective for data selection on much larger models.

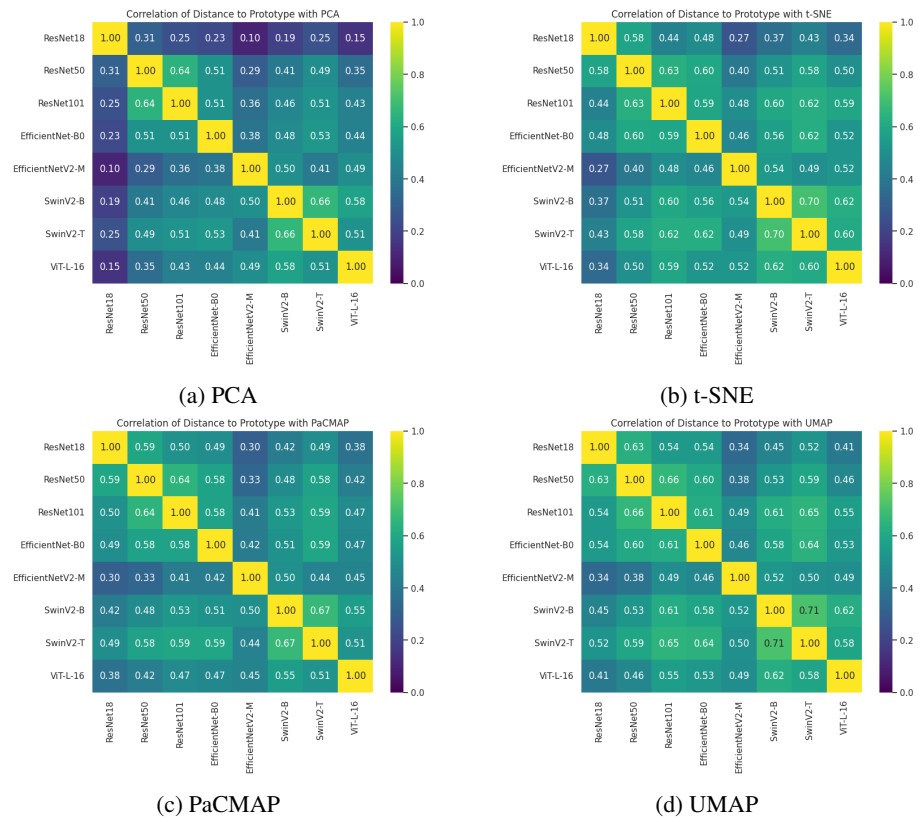

Figure 8: Correlation of per-sample distance to prototype across different architectures when applying different linear and non-linear manifold projection techniques.

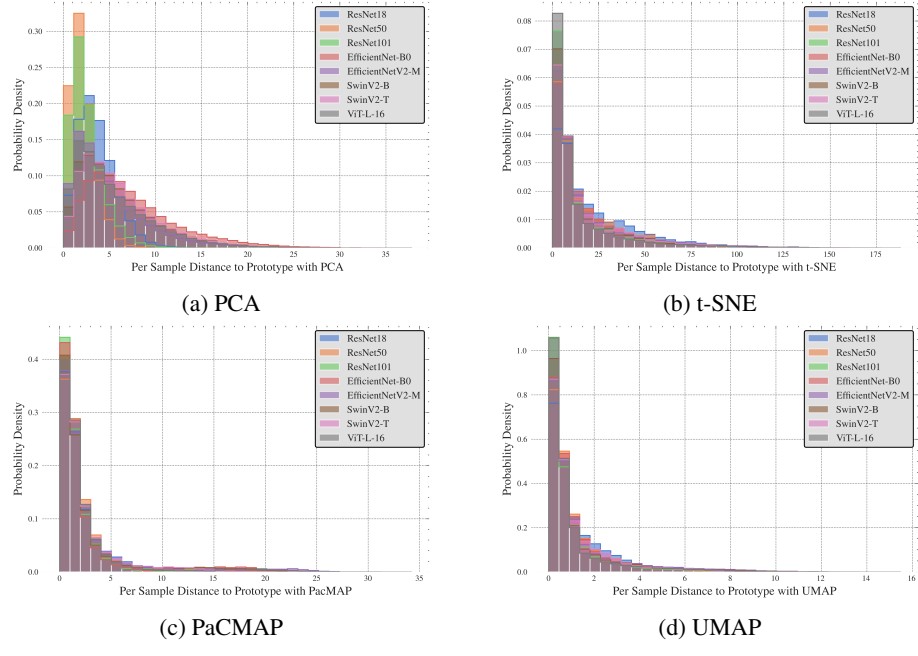

Figure 9: Distribution of per-sample distance to prototype across different architectures when applying different linear and non-linear manifold projection techniques.

## A.4 THE NUMBER OF PERSISTENCE OPTIMIZATION STEPS

We investigate the impact of the number of optimization steps for multi-parameter persistent homology (see Fig. 10). The number of required persistence optimization steps is inversely correlated with the final coreset size. When selecting a large coreset (e.g., at a 30% pruning rate), the selection process is robust, and even a few optimization steps (1-2) suffice to identify a high-quality subset. However, at high pruning rates (e.g., 90%), the task of distinguishing the most crucial samples becomes more sensitive, necessitating a greater number of optimization steps ($\geq 6$) to allow the point positions to converge and accurately reveal the most structurally important examples.

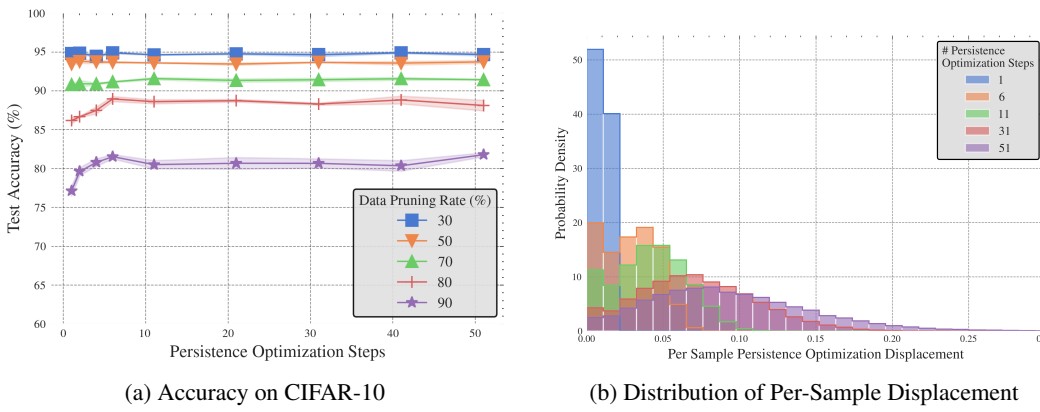

(a) Accuracy on CIFAR-10    (b) Distribution of Per-Sample Displacement

Figure 10: Smaller coresets have a lower margin for error, as the importance of each selected sample is magnified. Consequently, more optimization steps are needed to precisely distinguish the most critical samples. In contrast, larger coresets are more forgiving, requiring fewer steps to achieve a high-quality result.

## A.5 SENSITIVITY OF LOCAL PERSISTENCE ($\alpha$) AND GLOBAL DENSITY ($\beta$)

We investigate the impact of the hyperparameters $\alpha$ and $\beta$ from Eq. (5), which balance the influence of our global density and local persistence scores (see Fig. 11). Our analysis reveals that while the coreset quality is generally stable across a range of $(\alpha, \beta)$ values, a synergistic combination of both metrics consistently yields the best performance. Although using either density or persistence alone provides a reasonable baseline, combining them is particularly crucial at high pruning rates (e.g., 90%), where a balanced score improves accuracy by up to 5.4% over using either metric in isolation. This demonstrates that both global and local topology are vital for optimal selection and justifies our use of a fixed and balanced configuration set at $(50/50)$ across all experiments, minimizing the need for extensive hyperparameter tuning.

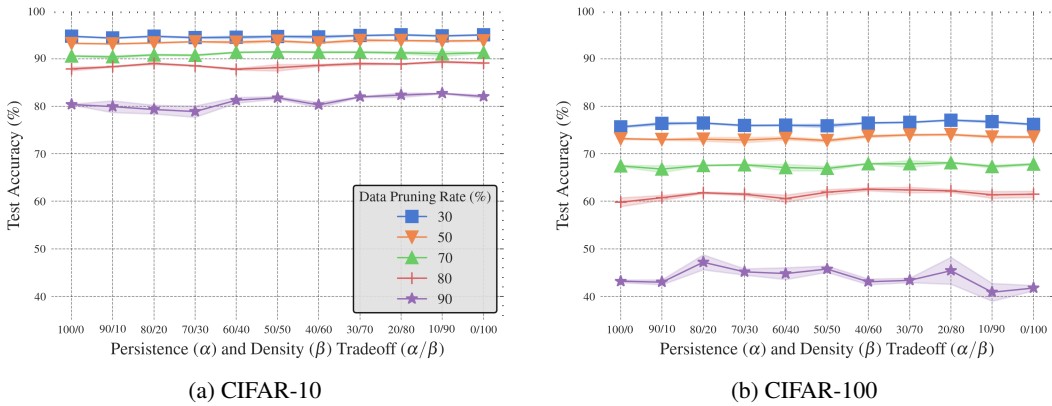

(a) CIFAR-10    (b) CIFAR-100

Figure 11: Topology hyperparameters across all ranges of data pruning rates for both (a) CIFAR-10 and (b) CIFAR-100.

## A.6 ILLUSTRATIVE EXAMPLE OF CORESET SELECTION

Qualitative visualization of TopoCore at various pruning rates (70%, 50%, 30%, 20% and 10%) for the "butterfly" class in CIFAR-100 (see Fig. 12). The visualization reveals a high variance in Persistence Scores within localized regions of the class manifold, demonstrating the method's sensitivity to fine-grained local structures and its ability to distinguish between nearby samples. Despite this focus on local complexity, the final coresets remain density-preserving, with their overall distribution closely matching that of the full dataset. This illustrates how TopoCore successfully balances the selection of topologically critical local samples with the preservation of the global data structure. The color-maps for all plots are normalized between 0 and 1.

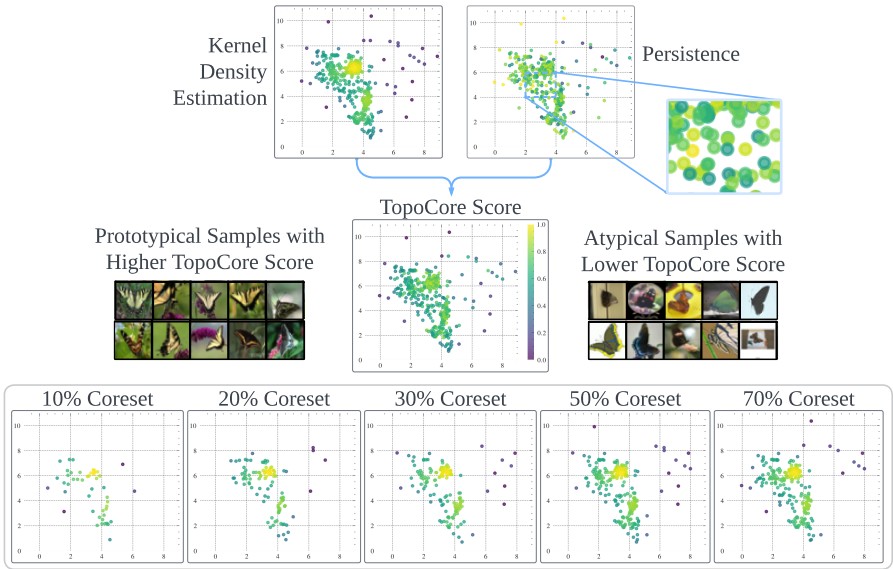

Figure 12: TopoCore for "butterfly" class in CIFAR-100.

## A.7 TRAINING-FREE PROXIES FOR AREA UNDER MARGIN (AUM)

We evaluate several training-free methods to serve as a proxy for the training-dependent Area Under the Margin (AUM) metric. These proxies identify potentially noisy samples using different geometric criteria, ranging from culling samples based on their *Distance* to the class prototype, to using an *Adjacent Distance* ratio to remove points closer to another class's manifold. Other heuristics include culling samples with the lowest *Density* score, or using our proposed *Neighborhood Label Purity Score (NLPS)*, which identifies points in mixed-label regions by calculating the fraction of same-label nearest neighbors (from 20 nearest neighbors). Our results show that NLPS provides the highest coreset accuracy among all training-free proxies, with its advantage being most pronounced at high data pruning rates. While it does not fully match the performance of using the true AUM, NLPS serves as a simple and effective proxy that does not require access to training dynamics.

Table 2: Comparison of training-dynamic free proxies for Area Under Margin (AUM) (Pleiss et al., 2020) on CIFAR-100. We see that Neighborhood Label Purity Score (NLPS) performs closest.

| Pruning Rate ($\rightarrow$) | 30% | 50% | 70% | 80% | 90% |
|---|---|---|---|---|---|
| Distance | 75.4±0.4 | 71.3±0.3 | 63.2±0.1 | 56.1±0.6 | 37.8±0.3 |
| Adjacent Distance | 75.5±0.3 | 71.8±0.2 | 62.8±1.0 | 57.1±0.3 | 38.1±1.5 |
| Density | 75.6±0.1 | 71.4±0.2 | 64.3±0.3 | 52.4±0.3 | 38.0±0.7 |
| NLPS | 75.6±0.2 | **71.9±0.2** | **65.3±0.4** | **56.7±0.4** | **41.6±1.0** |
| AUM | **75.9±0.4** | 72.8±0.3 | 66.9±0.5 | 61.9±0.6 | 45.7±0.7 |

## A.8 ON THE TRANSFERABILITY OF TOPOLOGICAL VS. EUCLIDEAN FEATURES

We provide a formal argument for the superior transferability of topological features derived from persistent homology over traditional Euclidean metrics across different neural network architectures (Papillon et al., 2025). We demonstrate that the stability guarantees inherent to persistent homology ensure that its output is robust to the geometric variations common between different network embeddings. Conversely, we show that Euclidean-based metrics, such as the distance to a class prototype, are inherently sensitive to these variations.

### PRELIMINARIES AND NOTATION

Let $X$ be the input data space and $Y = \{1, \ldots, K\}$ be the set of $K$ class labels. A neural network architecture is a function $f : X \to \mathbb{R}^n$ that maps input data to an $n$-dimensional embedding space. Let $f_A$ and $f_B$ denote two distinct network architectures (e.g., ResNet18 and ViT-L-16). The outputs of these networks for the entire dataset $X$ are the point clouds $X_A = f_A(X)$ and $X_B = f_B(X)$ in their respective embedding spaces. We equip these embedding spaces with the standard Euclidean metric, $d_E$.

**Definition A.3** (Bottleneck Distance). *The similarity between two persistence diagrams $Dgm_1$ and $Dgm_2$ is measured by the **bottleneck distance** $d_B(Dgm_1, Dgm_2)$, defined as the infimum over all bijections $\eta : Dgm_1 \to Dgm_2$ of the supremum of distances between matched points:*

$$d_B(Dgm_1, Dgm_2) = \inf_{\eta} \sup_{p \in Dgm_1} \|p - \eta(p)\|_\infty$$

*Points may also be matched to the diagonal. The $p$-Wasserstein distance $W_p$ is a related metric.*

**Definition A.4** (Gromov-Hausdorff Distance). *The distance between two metric spaces $(M_1, d_1)$ and $(M_2, d_2)$ is measured by the **Gromov-Hausdorff distance** $d_{GH}(M_1, M_2)$, which is the infimum of distances over all possible isometric embeddings into a common metric space. It quantifies the "metric dissimilarity" of two spaces.*

### INSTABILITY OF EUCLIDEAN DISTANCES TO PROTOTYPES

We now formalize the lack of such stability for Euclidean distances.

**Definition A.5** (Class Prototype and Distance Distribution). *For an embedding $f(X)$ and a class $k \in Y$, the class prototype (centroid) is $c_k = \frac{1}{|X_k|} \sum_{x \in X_k} f(x)$, where $X_k$ are the samples of class $k$. The set of distances to the prototype is $S_k(f) = \{d_E(f(x), c_k) \mid label(x) = k\}$. Let $P(S_k(f))$ be the probability distribution of these distances.*

**Proposition A.6** (Sensitivity to Scaling). *Let $f_A$ be a network embedding. Consider a new embedding $f_B$ defined by a simple isotropic scaling transformation, $f_B(x) = \alpha f_A(x)$ for some scalar $\alpha > 0, \alpha \neq 1$. Then the distribution of distances to the prototype is scaled accordingly: $P(S_k(f_B)) = \alpha P(S_k(f_A))$.*

*Proof.* The new class prototype $c'_k$ under the embedding $f_B$ is:

$$c'_k = \frac{1}{|X_k|} \sum_{x \in X_k} f_B(x) = \frac{1}{|X_k|} \sum_{x \in X_k} \alpha f_A(x) = \alpha \left( \frac{1}{|X_k|} \sum_{x \in X_k} f_A(x) \right) = \alpha c_k$$

The distance for any sample $x$ of class $k$ to the new prototype is:

$$\begin{aligned}
d_E(f_B(x), c'_k) &= d_E(\alpha f_A(x), \alpha c_k) \\
&= \|\alpha f_A(x) - \alpha c_k\|_2 \\
&= |\alpha| \cdot \|f_A(x) - c_k\|_2 = \alpha \cdot d_E(f_A(x), c_k)
\end{aligned}$$

Thus, every distance value in the set $S_k(f_A)$ is multiplied by $\alpha$ to obtain the set $S_k(f_B)$. The probability distribution of these distances is therefore a scaled version of the original. $\square$

STABILITY GUARANTEES FOR PERSISTENT HOMOLOGY

The transferability of persistence-based features is a direct consequence of the fundamental stability theorem of topological data analysis (Cohen-Steiner et al., 2005).

**Proposition A.7** (Invariance and Stability of Persistent Homology)**.**

1. **Isometry Invariance**: *Let $P \subset \mathbb{R}^n$ be a point cloud and $g : \mathbb{R}^n \to \mathbb{R}^n$ be a Euclidean isometry (translation, rotation, reflection). Then, the persistence diagram is unchanged: $Dgm(P) = Dgm(g(P))$.*

2. **Stability**: *Let $X_A$ and $X_B$ be two point clouds in $\mathbb{R}^n$. The bottleneck distance between their respective persistence diagrams is bounded by the Gromov-Hausdorff distance between their metric spaces:*

$$d_B(Dgm(X_A), Dgm(X_B)) \le d_{GH}((X_A, d_E), (X_B, d_E))$$

*Proof.* (1) An isometry $g$ preserves all pairwise Euclidean distances. Since the Vietoris-Rips filtration is constructed based solely on these distances, the filtration $\{VR(P, r)\}_{r \ge 0}$ is identical to $\{VR(g(P), r)\}_{r \ge 0}$. Applying the homology functor to identical filtrations yields identical persistence diagrams. (2) The proof is a cornerstone result in topological data analysis. It formalizes the intuition that if two spaces are metrically similar (a small $d_{GH}$), their topological features as captured by persistence homology must also be similar (a small $d_B$). □

A.9  ALL ACCURACY AND STANDARD DEVIATION TABLES

Table 3: A comparison of coreset performance under feature embedding noise on CIFAR-100. The plot shows mean accuracy vs. standard deviation (over 5 runs) for three geometric methods as noise ($\epsilon$) and pruning rates increase. While Moderate is precise (low variance) but inaccurate and D2 is more accurate but imprecise (high variance), TopoCore consistently delivers both high accuracy and high precision. The superiority of TopoCore is most evident at the highest pruning rates and noise levels, highlighting its robustness. Best accuracy and standard deviation values are shown in bold. If more than one method have the same value, the second one is underlined.

| Noise ($\to$) | $\epsilon \sim \mathcal{N}(0, 0.25\sigma)$ | | | | $\epsilon \sim \mathcal{N}(0, \sigma)$ | | | |
| --- | --- | --- | --- | --- | --- | --- | --- | --- |
| Pruning Rate ($\to$) | 50% | 70% | 80% | 90% | 50% | 70% | 80% | 90% |
| Moderate (Xia et al., 2023) | 71.1±0.2 | 63.7±0.4 | 56.0±0.6 | 33.2±0.9 | 70.4±**0.1** | 62.5±**0.4** | 54.6±0.6 | 33.9±**0.2** |
| D2 (Maharana et al., 2024) | 72.8±**0.1** | **68.5**±0.7 | **63.0**±0.5 | 44.4±1.5 | 73.0±**0.1** | 67.7±1.2 | 62.3±0.8 | 40.2±2.0 |
| **TopoCore** | **73.5**±0.3 | 67.8±**0.1** | 60.5±**0.2** | **45.4**±**0.8** | **73.3**±0.2 | **68.0**±0.5 | **62.7**±**0.3** | **45.5**±0.6 |
| Noise ($\to$) | $\epsilon \sim \mathcal{N}(0, 4\sigma)$ | | | | $\epsilon \sim \mathcal{N}(0, 8\sigma)$ | | | |
| Pruning Rate ($\to$) | 50% | 70% | 80% | 90% | 50% | 70% | 80% | 90% |
| Moderate (Xia et al., 2023) | 71.0±0.4 | 62.9±0.3 | 52.3±0.6 | 32.0±1.2 | 70.9±0.2 | 63.4±0.2 | 51.6±2.2 | 32.1±1.4 |
| D2 (Maharana et al., 2024) | 72.3±0.1 | **67.8**±1.1 | **62.1**±1.0 | 40.5±1.7 | 71.6±0.5 | 67.2±0.8 | 60.3±2.4 | 39.8±3.2 |
| **TopoCore** | **73.4**±**0.1** | 67.4±**0.2** | 61.7±**0.3** | **46.1**±**0.7** | **73.4**±0.2 | 67.2±**0.1** | **61.8**±**0.1** | **43.9**±**0.4** |

Table 4: Persistent Homology optimization steps on CIFAR-10. Optimal results are in bold, ties are underlined.

| | Pruning Rate (%) | | | | |
| --- | --- | --- | --- | --- | --- |
| # Steps | 30% | 50% | 70% | 80% | 90% |
| 1 | 94.8±0.1 | 93.5±0.2 | 90.8±0.2 | 86.2±0.4 | 77.1±0.6 |
| 2 | 94.9±0.1 | **93.8**±**0.3** | 90.9±0.5 | 86.7±0.1 | 79.6±0.5 |
| 4 | 94.5±0.2 | 93.7±0.2 | 90.9±0.1 | 87.5±0.5 | 80.8±0.5 |
| 6 | **94.9**±**0.1** | 93.7±0.1 | 91.2±0.1 | **89.0**±**0.3** | 81.5±0.3 |
| 11 | 94.6±0.1 | 93.6±0.1 | **91.6**±**0.2** | 88.6±0.3 | 80.5±0.5 |
| 21 | 94.8±0.1 | 93.5±0.2 | 91.3±0.2 | 88.7±0.2 | 80.7±0.8 |
| 31 | 94.7±0.2 | 93.7±0.1 | 91.4±0.2 | 88.3±0.1 | 80.7±0.6 |
| 41 | 94.9±0.1 | 93.6±0.2 | 91.6±0.2 | 88.8±0.5 | 80.4±0.7 |
| 51 | 94.7±0.2 | 93.7±0.2 | 91.4±0.1 | 88.1±0.7 | **81.8**±**0.3** |

Table 5: Accuracy and standard deviation across various coreset selection methods on CIFAR-10, CIFAR-100 and ImageNet-1K. New: ImageNet-1K results are now averaged over 3 runs.

| | | CIFAR-10 (ResNet-18) | | | | |
|---|---|---|---|---|---|---|
| | Pruning Rate ($\rightarrow$) | 30% | 50% | 70% | 80% | 90% |
| No Training Dynamics | Random | 94.5±0.1 | 93.5±0.1 | 90.8±0.2 | 86.6±0.3 | 76.7±0.9 |
| | Moderate (Xia et al., 2023) | 94.2±0.1 | 93.1±0.1 | 89.9±0.2 | 87.2±0.2 | 76.9±1.0 |
| | FDMat (Xiao et al., 2024) | 94.7±0.1 | 93.6±0.2 | 90.8±0.2 | 87.3±0.4 | 74.4±0.7 |
| | **TopoCore (w/ NLPS)** | **94.8±0.1** | **93.6±0.2** | 90.3±0.2 | **87.3±0.3** | **77.1±0.6** |
| With Training Dynamics | Moderate (w/ AUM) | 93.9±0.2 | 93.1±0.2 | 90.1±0.2 | 87.1±0.2 | 79.9±0.3 |
| | Forgetting (Toneva et al., 2019) | 94.5±0.2 | 92.6±0.1 | 89.8±0.2 | 85.6±0.3 | 67.6±0.4 |
| | Glister (Killamsetty et al., 2021b) | 94.4±0.2 | 93.8±0.2 | 90.8±0.4 | 85.1±0.6 | 66.8±1.3 |
| | LCMat-S (Shin et al., 2023) | 94.5±0.2 | 93.3±0.2 | 90.5±0.2 | 86.9±0.2 | 75.1±0.8 |
| | CCS (Zheng et al., 2023) | 95.5±0.1 | 94.8±0.2 | 93.0±0.2 | **90.7±0.2** | 81.9±0.7 |
| | D2 (Maharana et al., 2024) | **95.6±0.1** | **94.8±0.1** | **93.1±0.1** | 89.2±0.2 | 80.9±1.5 |
| | **TopoCore** | 94.7±0.2 | 93.7±0.2 | 91.6±0.1 | 88.7±0.4 | **82.1±0.3** |

| | | CIFAR-100 (ResNet-18) | | | | |
|---|---|---|---|---|---|---|
| | Pruning Rate ($\rightarrow$) | 30% | 50% | 70% | 80% | 90% |
| No Training Dynamics | Random | 75.3±0.2 | 71.6±0.1 | 63.7±0.5 | 55.9±1.0 | 34.0±1.1 |
| | Moderate (Xia et al., 2023) | 74.9±0.3 | 70.1±0.3 | 63.7±0.2 | 56.1±0.5 | 34.9±2.1 |
| | FDMat (Xiao et al., 2024) | 75.4±0.2 | 71.9±0.3 | 64.0±0.6 | 56.1±1.5 | 37.5±1.6 |
| | **TopoCore (w/ NLPS)** | **75.6±0.2** | **71.9±0.2** | **65.3±0.4** | **56.7±0.4** | **41.6±0.8** |
| With Training Dynamics | Moderate (w/ AUM) | 75.9±0.3 | 72.4±0.2 | 66.7±0.3 | 60.2±0.8 | 40.0±1.2 |
| | Forgetting (Toneva et al., 2019) | 74.8±0.2 | 67.2±0.9 | 50.6±0.7 | 32.3±0.9 | 24.3±1.4 |
| | Glister (Killamsetty et al., 2021b) | 75.8±0.3 | 70.7±0.7 | 66.1±1.2 | 54.7±1.6 | 38.4±1.7 |
| | LCMat-S (Shin et al., 2023) | 75.3±0.2 | 71.1±0.2 | 62.5±0.8 | 52.1±2.0 | 36.1±1.7 |
| | CCS (Zheng et al., 2023) | **76.9±0.3** | **73.8±0.3** | **67.8±0.7** | 60.7±0.6 | 45.2±2.4 |
| | D2 (Maharana et al., 2024) | 75.1±0.5 | 71.2±0.2 | 67.8±0.9 | 61.1±1.4 | 44.3±2.6 |
| | **TopoCore** | 75.9±0.4 | 72.8±0.3 | 66.9±0.5 | **61.9±0.6** | **45.8±0.7** |

| | | ImageNet-1K (ResNet-50) | | | | |
|---|---|---|---|---|---|---|
| | Pruning Rate ($\rightarrow$) | 30% | 50% | 70% | 80% | 90% |
| No Training Dynamics | Random | 69.8±0.5 | 68.4±0.5 | 65.1±0.4 | 61.9±0.5 | 52.5±0.6 |
| | Moderate (Xia et al., 2023) | 69.5±0.2 | 65.8±0.4 | 60.5±0.1 | 57.7±0.2 | 50.0±0.4 |
| | FDMat (Xiao et al., 2024) | **70.8±0.3** | 68.7±0.5 | 65.5±0.7 | 62.0±0.3 | 51.9±0.3 |
| | **TopoCore (w/ NLPS)** | 70.7±0.4 | **69.9±0.2** | **66.4±0.2** | **63.2±0.3** | **53.9±0.2** |
| With Training Dynamics | Moderate (w/ AUM) | 69.6±0.4 | 67.2±0.6 | 63.9±0.8 | 60.4±0.6 | 52.7±0.3 |
| | Forgetting (Toneva et al., 2019) | 69.9±0.2 | 66.8±0.6 | 60.2±0.5 | 59.1±0.4 | 50.0±0.5 |
| | Glister (Killamsetty et al., 2021b) | 66.3±0.4 | 63.5±0.3 | 59.3±0.5 | 56.5±0.3 | 49.3±0.8 |
| | LCMat-S (Shin et al., 2023) | 69.8±0.4 | 67.5±0.5 | 62.2±0.3 | 59.7±0.5 | 48.8±0.6 |
| | CCS (Zheng et al., 2023) | 70.1±0.5 | 69.1±0.3 | 65.7±0.3 | 62.6±0.6 | 55.2±0.7 |
| | D2 (Maharana et al., 2024) | 69.5±0.3 | 67.1±0.5 | 65.7±0.4 | 62.7±0.9 | 55.5±1.3 |
| | **TopoCore** | **70.8±0.2** | **69.5±0.2** | **66.2±0.1** | **63.1±0.3** | **56.1±0.2** |

Table 6: Transferability of features from many types of architectures to train a ResNet-18 model on CIFAR-100. Most models are taken from `torchvision` pretrained library which are finetuned from ImageNet-1K. We also look at the transferability of features from bigger OpenCLIP foundational models trained on the LAOIN-2b dataset (Schuhmann et al., 2022).

| Pruning Rate ($\rightarrow$) | 50% | | | 70% | | |
|---|---|---|---|---|---|---|
| | Moderate | D2 | **TopoCore** | Moderate | D2 | **TopoCore** |
| ResNet-18 (He et al., 2016) | 70.9±0.4 | 73.0±0.8 | 73.6±0.2 | 62.9±0.2 | 67.9±0.3 | 68.1±0.2 |
| ResNet-50 (He et al., 2016) | 71.1±0.1 | 73.0±0.2 | 73.7±0.2 | 63.3±0.4 | 67.7±0.4 | 68.0±0.1 |
| ResNet-101 (He et al., 2016) | 70.0±0.5 | 73.2±0.2 | 73.5±0.2 | 62.8±0.4 | 66.9±0.9 | 68.0±0.3 |
| EfficientNet-B0 (Tan & Le, 2019) | 71.6±0.3 | 73.2±0.2 | 72.9±0.2 | 62.9±0.1 | 67.5±0.4 | 67.5±0.2 |
| EfficientNetV2-M (Tan & Le, 2021) | 69.6±0.3 | 72.8±0.7 | 73.4±0.2 | 61.0±0.2 | 67.1±0.8 | 67.1±0.1 |
| SwinV2-T (Liu et al., 2022) | 70.5±0.1 | 73.6±0.1 | 73.6±0.2 | 60.4±0.6 | 66.9±1.0 | 67.6±0.1 |
| SwinV2-B (Liu et al., 2022) | 69.9±0.3 | 73.5±0.4 | 73.6±0.2 | 61.9±0.4 | 67.1±0.4 | 68.1±0.2 |
| ViT-L-16 (Dosovitskiy et al., 2021) | 69.8±0.3 | 73.3±0.3 | 73.6±0.1 | 61.5±0.3 | 67.2±0.6 | 68.0±0.1 |
| OpenCLIP ViT-L-14 (Radford et al., 2021) (Schuhmann et al., 2022) | 71.2±0.4 | 73.2±0.2 | 73.3±0.1 | 63.5±0.4 | 66.8±0.3 | 67.9±0.3 |
| OpenCLIP ViT-H-14 (Radford et al., 2021) (Schuhmann et al., 2022) | 70.9±0.3 | 73.0±0.2 | 73.1±0.4 | 62.4±0.3 | 66.6±1.3 | 67.7±0.2 |
| **Overall Average** | 70.6±0.3 | 73.2±0.3 | **73.4±0.2** | 62.3±0.3 | 67.2±0.7 | **67.8±0.2** |

| Pruning Rate ($\rightarrow$) | 80% | | | 90% | | |
|---|---|---|---|---|---|---|
| | Moderate | D2 | **TopoCore** | Moderate | D2 | **TopoCore** |
| ResNet-18 (He et al., 2016) | 54.8±0.2 | 60.3±1.9 | 60.2±0.2 | 33.8±0.8 | 42.5±1.9 | 43.4±0.4 |
| ResNet-50 (He et al., 2016) | 55.9±0.6 | 60.8±1.0 | 61.3±0.7 | 31.8±1.6 | 44.5±1.7 | 47.4±0.5 |
| ResNet-101 (He et al., 2016) | 54.5±0.4 | 60.4±0.2 | 60.0±0.6 | 35.9±0.6 | 41.7±2.6 | 43.2±1.3 |
| EfficientNet-B0 (Tan & Le, 2019) | 54.8±1.1 | 60.5±0.8 | 62.0±0.4 | 29.8±1.3 | 42.0±2.2 | 42.2±0.3 |
| EfficientNetV2-M (Tan & Le, 2021) | 53.1±0.5 | 60.2±0.4 | 60.1±1.6 | 33.3±0.8 | 41.3±1.9 | 44.4±1.7 |
| SwinV2-T (Liu et al., 2022) | 53.3±1.2 | 59.3±2.0 | 61.8±0.4 | 32.5±1.5 | 41.4±2.2 | 43.4±0.8 |
| SwinV2-B (Liu et al., 2022) | 53.5±0.2 | 60.2±1.5 | 61.1±0.6 | 35.7±0.3 | 42.8±2.9 | 42.7±1.6 |
| ViT-L-16 (Dosovitskiy et al., 2021) | 53.9±0.9 | 59.1±1.1 | 61.1±0.2 | 30.6±1.2 | 40.9±2.6 | 44.1±1.3 |
| OpenCLIP ViT-L-14 (Radford et al., 2021) (Schuhmann et al., 2022) | 53.9±0.8 | 60.4±1.0 | 61.7±0.8 | 35.0±1.5 | 40.7±2.9 | 42.5±1.6 |
| OpenCLIP ViT-H-14 (Radford et al., 2021) (Schuhmann et al., 2022) | 54.0±0.8 | 61.1±0.4 | 61.8±0.8 | 35.4±1.4 | 42.3±2.1 | 42.9±1.0 |
| **Overall Average** | 54.2±0.7 | 60.2±1.0 | **61.1±0.6** | 33.4±1.1 | 42.0±2.2 | **43.6±1.1** |

Table 7: Hyperparameters for local persistence ($\alpha$) and global density ($\beta$). While our fixed 50/50 split provides strong, stable performance, the results indicate that further accuracy gains are possible with task-specific tuning. We observe a trend where the optimal balance increasingly relies on the persistence score ($\alpha$) on more challenging datasets (e.g., CIFAR-100 vs. CIFAR-10) and at higher data pruning rates. Optimal results are in bold, ties are underlined.

| CIFAR-10 | Pruning Ratio (%) | | | | |
|---|---|---|---|---|---|
| $\alpha/\beta$ | 30% | 50% | 70% | 80% | 90% |
| 100/0 | 94.7±0.1 | 93.2±0.1 | 90.6±0.1 | 87.9±0.4 | 80.4±0.1 |
| 90/10 | 94.3±0.1 | 93.1±0.1 | 90.4±0.3 | 88.3±0.1 | 79.9±1.2 |
| 80/20 | 94.7±0.1 | 93.3±0.1 | 90.8±0.2 | 89.0±0.2 | 79.3±0.9 |
| 70/30 | 94.4±0.2 | 93.6±0.1 | 90.7±0.2 | 88.5±0.1 | 78.9±1.2 |
| 60/40 | 94.5±0.3 | 93.5±0.3 | 91.3±0.1 | 87.8±0.2 | 81.3±0.6 |
| 50/50 | 94.7±0.2 | 93.7±0.2 | **91.6±0.1** | 88.7±0.7 | 82.1±0.3 |
| 40/60 | 94.6±0.3 | 93.3±0.1 | 91.4±0.2 | 88.6±0.3 | 80.3±0.5 |
| 30/70 | 94.9±0.1 | **93.9±0.3** | 91.4±0.2 | 89.0±0.3 | 82.0±0.1 |
| 20/80 | **95.0±0.1** | 93.8±0.1 | 91.3±0.2 | 88.9±0.1 | 82.3±0.5 |
| 10/90 | 94.8±0.1 | 93.7±0.2 | 91.0±0.5 | **89.3±0.2** | **82.7±0.1** |
| 0/100 | 94.9±0.1 | 93.8±0.1 | 91.3±0.1 | 89.1±0.2 | 82.2±0.4 |

| CIFAR-100 | Pruning Ratio (%) | | | | |
|---|---|---|---|---|---|
| $\alpha/\beta$ | 30% | 50% | 70% | 80% | 90% |
| 100/0 | 75.7±0.2 | 73.2±0.2 | 67.4±0.2 | 59.8±0.9 | 43.2±0.3 |
| 90/10 | 76.4±0.3 | 73.0±0.1 | 66.8±0.7 | 60.8±0.5 | 43.0±0.5 |
| 80/20 | 76.5±0.1 | 73.1±0.4 | 67.5±0.1 | 61.8±0.2 | **47.2±1.6** |
| 70/30 | 76.0±0.2 | 72.9±0.6 | 67.7±0.2 | 61.5±0.3 | 45.2±0.7 |
| 60/40 | 76.0±0.2 | 73.3±0.3 | 67.1±0.7 | 60.6±0.8 | 44.8±1.2 |
| 50/50 | 75.9±0.4 | 72.8±0.3 | 66.9±0.5 | 61.9±0.6 | 45.8±0.7 |
| 40/60 | 76.5±0.1 | 73.7±0.3 | 67.9±0.1 | **62.5±0.3** | 43.1±0.6 |
| 30/70 | 76.6±0.1 | **74.0±0.2** | 67.9±0.7 | 62.4±0.6 | 43.4±0.5 |
| 20/80 | **77.1±0.1** | 74.0±0.2 | **68.1±0.1** | 62.2±0.3 | 45.4±2.8 |
| 10/90 | 76.7±0.2 | 73.6±0.3 | 67.3±0.3 | 61.4±0.7 | 40.9±1.9 |
| 0/100 | 76.2±0.1 | 73.5±0.2 | 67.8±0.2 | 61.5±0.7 | 41.8±0.6 |

## A.10 TRAINING AND TOPOLOGICAL HYPERPARAMETERS

Table 8: Training and topological hyperparameters.

| Section | Hyperparameter | CIFAR-10 CIFAR-100 | ImageNet |
|---|---|---|---|
| Training (`DeepCore`) | Epochs | 200 | 60 |
| | Batch Size | 256 | 128 |
| | Optimizer | SGD | SGD |
| | Momentum | 0.9 | 0.9 |
| | Learning Rate | 1e-1 | 1e-1 |
| | Weight Decay | 5e-4 | 5e-4 |
| | Scheduler | CosineAnnealing | CosineAnnealing |
| Global Manifold Projection (`UMAP`) | Number Neighbors | 15 | 15 |
| | Minimum Distance | 0.1 | 0.1 |
| | Metric | Cosine | Cosine |
| | Dimensions | 2 | 2 |
| Kernel Density Estimation (`sklearn`) | Bandwidth | 0.4 | 0.4 |
| Local Persistent Homology (`multipers`) | Theta (Density Bandwidth) | 0.4 | 0.4 |
| | Function/Kernel | Gaussian | Gaussian |
| | Complex | Weak-Delaunay | Weak-Delaunay |
| | Homology Degree | 1 | 1 |
| | Optimization Steps | 6 | 6 |
| Topology Score | Global Density ($\alpha$) | 0.5 | 0.5 |
| | Local Persistence ($\beta$) | 0.5 | 0.5 |

Table 9: Dataset mislabel ratios. Similar to those in Zheng et al. (2023).

| | Mislabel Ratio (%) | | |
|---|---|---|---|
| Pruning Rate | CIFAR-10 | CIFAR-100 | ImageNet-1k |
| 30% | 0% | 10% | 0% |
| 50% | 0% | 20% | 10% |
| 70% | 10% | 20% | 20% |
| 80% | 10% | 40% | 20% |
| 90% | 30% | 50% | 30% |

## A.11 LLM USAGE POLICY

In accordance with ICLR's Large Language Model (LLM) Usage Policy, we disclose that the LLM Gemini 2.5 was used for the sole purpose of refining and improving the written clarity in this paper.

## A.12 COPYRIGHTS

Datasets: CIFAR-10 (unknown), CIFAR-100 (unknown), ImageNet (CC BY-NC 4.0). Libraries: Multipers (MIT License, Copyright (c) 2023 David Loiseaux), DeepCore (MIT License, Copyright (c) 2023 Zhao, Bo), UMAP (BSD 3-Clause License, Copyright (c) 2017, Leland McInnes)

