# OpenReview forum: "TopoCore: Unifying Topology Manifolds and Persistent Homology for Data Pruning"
_ICLR.cc/2026/Conference — Submitted to ICLR 2026_

### Official Review · Reviewer_hXms · 2025-10-17

**Soundness:** 3
**Presentation:** 4
**Contribution:** 3
**Rating:** 6
**Confidence:** 3

**Summary:**

The paper addresses the coreset selection problem: choosing a small subset of training data that maintains nearly the same model performance as the full dataset. It introduces a training-free approach that operates on frozen embeddings, viewing the dataset as a point cloud and using both global manifold density and local topological persistence to identify samples essential to the intrinsic structure. On benchmark image datasets, the method consistently achieves higher retention and lower variance than geometric baselines, especially under high pruning, showing stability and robustness across architectures, though its experiments are limited to vision tasks and lack compute analysis.

**Strengths:**

1. **Originality and clarity.** A training free coreset that combines manifold density with local topological persistence is a fresh, well motivated idea and the method is clearly described for replication.

2. **Strong empirical results.** Consistently high retention and lower variance than geometric baselines, especially at high pruning, plus sensible ablations on mixing weights and optimization depth.

3. **Practical robustness.** Works across multiple backbones with good transfer and noise robustness, indicating the selection signal is less model dependent than distance based methods.

**Weaknesses:**

1. **Limited scope.** Evaluation is restricted to vision benchmarks; no NLP or other modalities are tested, which weakens generality claims.

2. **Dependence on embeddings and projection**. Results hinge on the quality of frozen features and the chosen manifold projector, with limited guidance on hyperparameters or stability across settings.

3.  **Unclear computational costs.** No clear wall clock, memory, or scaling analysis for kNN construction and persistence steps, so the cost–accuracy tradeoff is unclear.

4. **Quantitative evidence is incomplete.** The paper’s claims, like *“up to 4× better precision” and improved proxy-to-target transfer*, are not consistently backed by tables, and the sensitivity of results to k-NN and manifold-projection settings remains largely unexplored.

**Questions:**

1. How sensitive are results to the manifold projection choice and its hyperparameters?

2. Please provide runtime and memory comparisons vs baselines on CIFAR and ImageNet to clarify scalability.

3. Do you have any non-vision results to support generality, for example ANLI or IMDB with a frozen RoBERTa encoder (D2 paper).

I like this paper and find the direction promising. I am at marginal accept and am willing to increase my score if the authors address my concerns in the rebuttal with concrete evidence.

---

> ### Author Response · Authors · 2025-11-20
>
> Thank you hXms for your thoughtful review. We particularly appreciate your understanding of the key strengths of our approach and for finding this direction promising. We hope that the following response answers all of the points you brought up regarding our work:
>
> ---
> ### Response to Weaknesses/Questions
>
> > W1/Q3: Evaluation is restricted to vision benchmarks… Do you have any non-vision results?
>
> - That is an excellent point. We originally intended to include NLP benchmarks, specifically the ANLI and IMDB evaluations as performed in D2 [1]. However the [open-sourced D2 repository](https://github.com/adymaharana/d2pruning) _excludes_ NLP evaluation code, and our attempts to reimplement their setup from scratch failed to reproduce the reported results. Lacking a verifiable baseline, we prioritized vision benchmarks where methods are publicly available to ensure rigorous, reproducible science. We agree that extending this to NLP is an important future direction.
> - We hypothesize our topology-based approach is fundamentally task-agnostic. Our method operates on the intrinsic geometry of frozen embeddings, a property independent of the source modality. This is supported by recent literature utilizing TDA to analyze multi-modal embedding spaces like CLIP [2, 3] and the "reasoning manifold" of LLM's [4].
> - Empirically, Table 6 (Appendix) offers preliminary evidence supporting the crosss-modality transfer. TopoCore successfully uses multi-modal OpenCLIP embeddings trained on LAION-5B [5] to select high-fidelity coresets for CIFAR-100, demonstrating that topological structure is transferable across modalities.
>
> > W2/Q1: Results hinge on the quality of frozen features and the chosen manifold projector… How sensitive are results to the manifold projection choice and its hyperparameters?
>
> - This is an excellent suggestion. Please see Section 3 (Sensitivity to Topology-based hyperparameters) of the Global Response for the detailed results and analysis.
>
> > W3/Q2: Unclear computational costs… Please provide runtime and memory comparisons vs. baselines to clarify scalability.
>
> - We agree that assessing practical scalability is crucial. Please refer to Section 2 (Computational complexity and wall-clock analysis) in the Global Response, where we provide a detailed Big-O complexity derivation and empirical runtime benchmarks against geometric baselines.
>
> > W4.1: Claims like *“up to 4x better precision”* are not consistently backed by tables.
>
> - We appreciate that you highlighted this. The claim refers specifically to the statistical stability (standard deviation) of our method compared to some of the most competitive baselines (e.g., D2 [1] and CCS [6])
> - As shown in Table 5 (Appendix), at the 90% pruning rate on CIFAR-100, TopoCore achieves standard deviation of 0.7% whereas D2 exhibits 2.6%, an improvement of roughly 3.7$\times$. We also see that at 90% pruning rate for CIFAR-10, TopoCore 8$\times$ standard deviation compared to D2
> - **Revision:** We agree this claim requires clear context (especially in the Abstract) requires better context. We will explicitly contextualize this claim in Section 4.2 (Results) with a direct reference to Table 5.
>
> > W4.2  Claims like *improved proxy-to-target transfer* are not consistently backed by tables.
>
> - This claim is substantiated by two sets of experiments.
> - *(1) Diverse Proxies $\rightarrow$ Fixed Target (Appendix Table 6)*: As detailed in Section 4.3, we evaluated the transferability of embeddings from a wide range of proxy networks (ResNet, EfficientNet, Swin, ViT, OpenCLIP) to train a fixed ResNet-18 target. TopoCore consistently yielded higher accuracy and lower variance compared to geometric baselines (Moderate [7], D2 [1]) across _all_ tested proxy architectures.
> - *(2) Fixed Proxies $\rightarrow$ Diverse Targets (New)*: To further validate generalizability, we performed an additional ablation (see response to Sq9q) using a single proxy (ResNet) to select coresets for training a wide range of target models (EfficientNet, Swin Transformers). These results confirm that TopoCore selects highly transferable coresets regardless of the transfer direction.
>
> ---
> ### References
>
> [1] Maharana et al. “D2 pruning: Message passing for balancing diversity & difficulty in data pruning.” *ICLR*, 2024 \
> [2] Rahim et al. "Topological Perspectives on Optimal Multimodal Embedding Spaces." *ArXiv*, 2024 \
> [3] You et al. "Topological Alignment of Shared Vision-Language Embedding Space." *ArXiv*, 2025 \
> [4] Li et al. "REMA: A Unified Reasoning Manifold Framework for Interpreting Large Language Model." *ArXiv*, 2025 \
> [5] Schuhmann et al. “LAION-5b: An open large-scale dataset for training next generation image-text models”. *NeurIPS D&B Track*, 2022 \
> [6] Zheng et al. "Coverage-centric coreset selection for high pruning rates". *ICLR*, 2023 \
> [7] Xia et al. "Moderate coreset: A universal method of data selection for real-world data-efficient deep learning". *ICLR*, 2023

---

> ### Comment · Reviewer_hXms · 2025-11-27
>
> After reading the rebuttal, new experiments, and the revised draft, I am positively updated on this work. The authors address several of my concerns (UMAP sensitivity and alternatives, quantitative backing for stability, and proxy->target transfer), and I find the overall direction promising. However, the new wall–clock analysis also makes it clear that TopoCore is currently much slower than strong geometric baselines (about an order of magnitude slower than D2 on CIFAR-100), and the evaluation remains confined to vision benchmarks with some methodological details still harder to follow than ideal. For these reasons, I will keep my score at 6, but I would be happy to see this paper accepted.

---

> > ### Author Response · Authors · 2025-12-02
> >
> > **Note regarding the Rebuttal Period:** We acknowledge that the discussion period has officially closed early due to the OpenReview leak. However, for the sake of completeness we provide the response we would have submitted below.
> >
> > ---
> >
> > Dear hXMs,
> >
> > We sincerely thank you for the positive update and for recognizing the value of our revisions regarding sensitivity, stability, and transferability. We appreciate your support for the paper's acceptance.
> >
> > > Comment: Wall-clock analysis makes it clear that TopoCore is currently much slower than strong geometric baselines.
> >
> > - We acknowledge the current gap compared to D2. However, our low hardware utilization (16% vs. 82%) confirms this is an implementation bottleneck (lack of multi-processing or GPU support in `multipers`) rather than an algorithmic one. With similar utilization, TopoCore would be only 1.7$\times$ slower than D2, significantly narrowing the perceived 8.9$\times$ gap.
> >
> > > Comment: Evaluation remains confined to vision benchmarks.
> >
> > - We restricted evaluation to vision to ensure rigorous reproducibility, as the NLP code for baselines like D2 is unavailable, and none of the other baselines (in Table 1) include NLP comparisons. However, our results with OpenCLIP (Appendix Table 6) demonstrate that the topological structure we probe is transferable across modalities, supporting our hypothesis that the method is task-agnostic. We agree that extending coreset evaluations to NLP benchmarks is a vital future direction for the field.
> >
> > > Comment: Methodological details harder to follow than ideal.
> >
> > - To address clarity, we have added a dedicated End-to-End Pipeline section (Section 3.5) to explicitly detail the sequential workflow of Scoring, Filtering, and Selection. We hope this further clarifies our methodology.

---

### Official Review · Reviewer_y5KN · 2025-10-23

**Soundness:** 2
**Presentation:** 2
**Contribution:** 2
**Rating:** 2
**Confidence:** 3

**Summary:**

The paper introduces TopoCore, a method for coreset selection. It is a combination of dimensionality reduction, non-parametric density estimation and persistent homology. Then, coresets are used for further training of ResNet-18.
Experimental results show that the proposed method slightly outperforms baseline.

**Strengths:**

The paper proposes a new approach to coreset selection.
This is one of the few applications of multipersistence to deep learning.

**Weaknesses:**

1) The paper is hard to understand. Some notions like "Hilbert decomposition signed measure" are not defined.
2) Some details of the method are missing (see Questions)
3) The difference is no statistically significant w.r.t. baselines in many cases (Table 5 in Appendix).
Please include statistical tests to validate significance.
4) Improvements over Random selection is quite small. I doubt that the method is of practical importance.
5)  A relevant publications is missing:

Trofimov, I., Cherniavskii, D., Tulchinskii, E., Balabin, N., Burnaev, E., & Barannikov, S. (2023). Learning topology-preserving data representations. arXiv preprint arXiv:2302.00136.

**Questions:**

1) As far as I understood, persistence scores are calculated for every class separately.  Are they summed next?
2) The optimization of L_{pers} can naturally lead to a degenerate solution, like points very far from each other, which maximizes persistence. How do you handle it?
3) Is TopologyScore maximized or minimized or minimized?
4) In Table 1, why TopoCore exhibits different metrics in "no training dynamics" and "with training dynamics" blocks?
I assume that the difference must be only in baselines.
5) Some important details are hard to understand from the paper. How L_{proj} is optimized? Together with TopologyScore or not?
How the coreset is selected? Is should be a subset of a dataset, but I can't find details.
Where are similarities p_{ij} are taken from? etc.

---

> ### Author Response · Authors · 2025-11-20
>
> Thank you y5KN for your time and feedback. We appreciate the opportunity to clarify the technical details of our method and demonstrate its practical significance. We hope the following responses resolve the questions raised:
>
> ---
> ### Response to Questions
>
> > Q1: Persistence scores are calculated for every class separately. Are they summed next?
>
> - No, scores are calculated for each individual sample $y_i$ within its specific class manifold $Y_c$. The final `TopologyScore` for a sample is the weighted sum of its own _local_ persistence ($Score_{pers}$) and _global_ density ($Score_{dens}$), used to rank samples within that class.
>
> > Q2: Optimization of $L_{pers}$ can naturally lead to a degenerate solution, like points very far away from each other... How do you handle this?
>
> - While valid for 1-parameter optimization, our 2-parameter descriptor incorporates both (1) the Vietoris-Rips filtration (distance) and (2) a Kernel Density Estimator (density). The signed measure $\mu^{Hil}$ explicitly encodes density, if points migrate too far from the high-density manifold, this negatively impacts the objective. This explicitly anchors points to the underlying structure (see Figure 2, middle), preventing degeneracy or scattering. Please see Section 3.2 ("... the optimization *enhances topological stability while preserving the original density of the class manifold*, as the density is recomputed ...")
>
> > Q3: Is TopologyScore maximized or minimized?
>
> - The `TopologyScore` is a *ranking metric*, not a loss function to be minimized. A high score indicates a sample that is both representative of the global class manifold (high density) and topologically critical (high persistence/structural integrity). Please refer to Appendix A.6, which visualizes how these components combine to identify "prototypical" samples.
>
> > Q4: Why does TopoCore have two separate metrics (“no training dynamics" and “with training dynamics”)?
>
> - As described in Section 4.1, we present two variants of our framework.
> - (1) *"No training dynamics"* uses our completely training-free proxy for identifying mislabeled/difficult samples (Section 3.4).
> - (2) *"With training dynamics"* incorporates the standard Area Under Margin (AUM) metric which requires training, to benchmark fairly against other coreset methods that rely on training dynamics (like CCS or D2).
>
> > Q5: Details which are hard to understand … How is $L_{proj}$ optimized? … How is the coreset selected?
>
> - This method is sequential (visualized in Figure 2).
> - (1) We optimize $L_{proj}$ using UMAP to project data into a low-dimensional manifold.
> - (2) We optimize $L_{pers}$ on this low-dimensional manifold to calculate per-sample persistence scores.
> - (3) We compute a final `TopologyScore` (combining persistence and low-dimensional manifold density) and perform stratified sampling to construct the coreset.
>
> ---
> ### Response to Weaknesses
>
> > W1: The paper is hard to understand. Some notions like "Hilbert decomposition signed measure" are not defined.
>
> - We appreciate you highlighting this. While our original submission included the description: _"This descriptor represents the persistence diagram as a finite collection of positive point masses (representing feature births) and negative point masses (representing feature deaths) in the parameter space of (distance, density),"_ we acknowledge that this text was disconnected from the formal terminology, leading to ambiguity.
> - **Revision:** We will explicitly couple this definition with the term Hilbert decomposition signed measure in Section 3.2 and add the citation by Loiseaux et al. [1] to ensure clarity.
>
> > W3/W4: Statistical significance w.r.t baseline … and improvement over baselines such as Random selection … I doubt that the method is of practical importance.
>
> - We appreciate your perspective. However, we note that the absolute accuracy gains in the coreset literature are often tight (typically $<1\%$), TopoCore offers distinct practical advantages that baselines do not:
> - **(1) Stability:** We demonstrate statistical reliability ($4\times$ lower variance than D2, Table 5) and robustness to noisy features (maintaining quality even when features are perturbed by $8\times$ std dev, Section 4.4), unlike geometric baselines.
> - **(2) Efficiency:** We operate on frozen embeddings (minutes) rather than requiring costly proxy model training (hours).
> - **(3) Transferability:** Our coresets transfer effectively across diverse architectures (Table 6) and target models (see response to Sq9q), a critical feature for production pipelines where retraining selection models is infeasible.
> - We believe establishing a practical use case for differential topology to probe embedding spaces contributes significant novelty and utility to the ICLR community.
>
> ---
> ### References
>
> [1] D. Loiseaux et al. "Stable vectorization of multiparameter persistent homology using signed barcodes as measures". *NeurIPS*, 2023

---

> > ### Comment · Reviewer_y5KN · 2025-11-27
> > **Response**
> >
> > Thank you for the answer, many of my questions are resolved.
> >
> > 1) It seems that the Persistence Score is high for outliers (see Appendix A.6). Is it intended behaviour of the score?
> > To my understanding, for CoreSets one should pick prototypical objects, not outliers.
> > 2) If you minimize the total persistence score (3), it seems that Y_c will stick together in one cluster.
> > (Line 280: The optimization seeks a new point configuration Y'c that minimizes this loss).
> > The illustrations in block "Local Differentiable Persistent Homology Optimization" in Fig.2 doesn't help since point clouds are visually non-distinguishable.
> > The formal algorithm will help a lot to understand the proposed method.
> > How can persistence increase at t=50 if you minimized (3) ?
> > 3) How NLPS score is incorporated into TopologyScore ? If it has any hyperparameters, how they are tuned?
> >
> > I acknowledge empirical performance of your method, especially with high pruning rates. The confusion in the presentation of your method is a obstacle.

---

> > > ### Author Response · Authors · 2025-12-02
> > >
> > > **Note regarding the Rebuttal Period:** We acknowledge that the discussion period has officially closed early due to the OpenReview leak. However, for the sake of completeness and to address the final questions raised by y5KN, we provide the response we would have submitted below.
> > >
> > > ---
> > >
> > > Dear y5KN,
> > >
> > > We are glad to hear that our previous responses addressed most of your concerns, and we sincerely appreciate you raising your overall rating from 2 $\rightarrow$ 4 and presentation score from 2 $\rightarrow$ 3. We also value your acknowledgment of the method's strong empirical performance. Below, we address your remaining questions. Additionally, we highlight that the workflow mechanics discussed here are fully implemented in the anonymous code repository provided to the reviewers and area chairs.
> > >
> > > > Q1: Persistence is high for outliers (see Appendix A.6)? ... Coresets should pick prototypical samples, not outliers.
> > >
> > > - This is a crucial distinction. Persistence alone is indeed insufficient to distinguish between "noise" outliers and "structural" boundary points.
> > > - The *Persistence Score* ($Score_{pers}$) measures _local_ structural significance relative to immediate neighbors (e.g., points maintaining a loop or bridge). This is exactly why we incorporate the *Global Density Score* ($Score_{dens}$). Density acts as a global prior for "prototypicality."
> > > - By summing these scores, true outliers (which have very low density) receive a low final `TopologyScore`, even if their persistence is high. Conversely, the "prototypical" samples we select are those that reside in high-density regions _and_ contribute significantly to the local manifold structure (high persistence). This interplay is visualized in the color gradient of Figure 12 (specifically the colormap scale in the middle section`TopoCore Score`).
> > >
> > >
> > > > Q2.1: In Figure 2, point clouds are visually non-distinguishable after optimization seeks a new point configuration Y'c.
> > >
> > > - We apologize if the static visualization makes the movement difficult to perceive. The optimization does not aim to radically reshape the class manifold, but rather to perform subtle, local (or micro-level) displacements.
> > > - Because we optimize for $H_1$ (homology degree 1), the points shift slightly to "open up" or clarify loops within the class manifold.
> > > - To visualize the difference between `t=0` and `t=50` please zoom in a bit more to see the changes in the class manifold point cloud.
> > >
> > > > Q2.2: In Figure 2 ... How can persistence increase at t=50 if you minimized (3)
> > >
> > > - Thank you for bringing up this clarification point. Our optimization objective is to _maximize_ the topological signal (lifespan of features) to identify critical points.
> > > - The caption "Samples exhibit greater persistence" refers to the *persistence lifecycle* (the time difference between the birth and death of simplices). The optimization displaces points such that the underlying topological features become more robust (longer-lived).
> > > - **Revision:** In the final manuscript, we will update the Figure 2 text to be precise: _"Samples exhibit longer persistence lifecycles (increased duration between birth and death of simplices)."_
> > >
> > > > Q3: How NLPS score is incorporated into TopologyScore ? If it has any hyperparameters, how they are tuned?
> > >
> > > - NLPS (Neighborhood Label Purity Score) or the original AUM (Area Under Margin) scores act as filters during the coreset selection phase, not as a term in the weighted score.
> > > - The process is sequential. (1) **Scoring:** We calculate `TopologyScore` for every sample in a dataset. (2) **Filtering:** We apply NLPS (or AUM) to identify samples with high label noise, those which are most likely mislabeled. (3) **Selection:** From the *remaining (clean) dataset*, we perform selection based on the `TopologyScore`.
> > > - We utilize a fixed neighborhood size of `n_neighbors=20`. As shown in our comparison of training-free noise proxies in Appendix A.7, this setting provides robust noise rejection without the need for further dataset-specific tuning.
> > > - This filtering strategy aligns with standard protocols established by CCS [1] and adopted by baselines like D2 [2] to ensure coreset purity.
> > >
> > > > Comment: The presentation of your method is a obstacle.
> > > - We appreciate this feedback and acknowledge that the initial presentation of the end-to-end workflow could have been clearer.
> > > - **Revision:** To address this, we have added a dedicated subsection (3.5 Topological Stratification and Coreset Construction). This new section clarifies the sequential nature of the process into three distinct phases: (1) Dual-scale topological scoring, (2) Mislabel sample filtering, and (3) Topological selection, ensuring the methodological flow is transparent and easy to follow.
> > >
> > > ---
> > > ### References
> > >
> > > [1] Zheng et al. "Coverage-centric coreset selection for high pruning rates". _ICLR_, 2023 \
> > > [2] Maharana et al. “D2 pruning: Message passing for balancing diversity & difficulty in data pruning.” _ICLR_, 2024

---

### Official Review · Reviewer_Sq9q · 2025-10-30

**Soundness:** 2
**Presentation:** 3
**Contribution:** 2
**Rating:** 4
**Confidence:** 3

**Summary:**

The paper addresses the problem of coreset selection, i.e. a small representative subset of a large dataset that minimizes the degradation in model performance and allows for faster training and reduced storage. Although existing geometry-based methods do not require an expensive training, they rely on extrinsic metrics that make them sensitive to variations in feature embeddings. The authors propose TopoCore, a two-stage method for coreset selection that utilizes topology to accurately approximate the underlying manifold of the data. To preserve the global structure, during the first stage, feature embeddings of deep neural network are projected onto a low-dimensional manifold with UMAP. To preserve the local structure, during the second stage topological persistence of points is maximized independently for each class. The coreset selection is based on the TopologyScore that combines Density Score, reflecting global representativeness, and Persistence Score, reflecting local topological complexity. The empirical evaluation includes comparison with several baseline methods in both training-based and training-free scenarios, analysis of method’s performance when feature embedding model is varied. The authors also analyze the TopoCore robustness to the noise injected into feature embeddings.

**Strengths:**

- The paper proposes a novel topology-based view on the problem of coreset selection that leverages the geometric methods.
- Experimental results demonstrate that TopoCore outperforms benchmark methods, especially at high pruning rate and on more complex datasets.
- TopoCore is more robust to noise in the feature space, especially at the higher pruning rates (70-90%).
- TopoCore provides better results across a wide range of embedding model choice.

**Weaknesses:**

- Although the paper provides some evidence for the choice of UMAP, a more recent works [1][2][3], which were shown to outperform UMAP with better preservation of data topology, are not considered for comparison and/or improvement of TopoCore.
- Experimental part is limited. As far as I understand, the experiments focus on the test accuracy of the ResNet-family models (ResNet-18, ResNet-50) for different pruning rates and embedding models. The evaluation lacks results for more recent architectures, for example, transformers, and estimation of other properties such as quality of transfer learning / domain adaptation.
- The paper does not provide any estimate on the computational cost of the proposed procedure. Is TopoCore more computationally intensive than the benchmark methods?

Minor: The notion of prototype is often used in the main text but formal definition is given only in the appendix.

[1] M. Moor et al. Topological autoencoders. ICLR, 2020.
[2] I. Trofimov et al. Learning topology-preserving data representations. ICLR, 2023.
[3] E. Tulchinskii et al. RTD-Lite: scalable topological analysis for comparing weighted graphs in learning tasks. AISTATS, 2025.

**Questions:**

Please, see weaknesses.

---

> ### Author Response · Authors · 2025-11-20
>
> We are grateful for your insightful feedback and careful consideration of our work Sq9q. Your constructive feedback provides both encouragement and practical guidance for strengthening this work. We are eager to respond to your questions and comments:
>
> ---
>
> ### Response to Weaknesses/Questions
>
>
> > W1: Works [1, 2, 3] … with better preservation of data topology, are not considered for comparison.
>
> - We agree this is an important comparison. Please see Section 1 (Topological autoencoders as a plugin for UMAP) in the Global Response.
>
> > W2: The evaluation lacks results for more recent architectures, for example, transformers, and estimation of other properties such as quality of transfer learning / domain adaptation.
>
> - We appreciate this suggestion. While our original submission evaluated various frozen embeddings (ResNet, EfficientNet, Swin, ViT) to train a fixed ResNet-18 target (Section 4.3), we agree that the inverse, using fixed embeddings to train diverse target architectures, is equally important.
> - Below, we demonstrate the transferability of TopoCore by using a standard ResNet proxy to select coresets for training EfficientNet and Swin Transformer models on CIFAR-100 and ImageNet-1K
>
> **Table 1: Cross-architecture transfer on CIFAR-100.** We compare the "Oracle" performance (coreset selected using the target model's own embeddings) against TopoCore transfer (coreset selected using ResNet-18 embeddings). Results are averaged over 3 runs.
>
> | **Target Model**           | **Pruning** | Oracle Acc.    | **TopoCore (ResNet-18)** | Difference |
> | -------------------------- | ----------- | -------------- | ------------------------ | ---------- |
> | **ResNet-50** (25.6M)      | 50%         | 73.0 $\pm$ 0.1 | 72.3 $\pm$ 1.0           | **-0.7**   |
> |                            | 70%         | 65.5 $\pm$ 0.5 | 63.8 $\pm$ 1.3           | **-1.7**   |
> |                            | 80%         | 57.0 $\pm$ 0.6 | 56.1 $\pm$ 0.7           | **-0.9**   |
> |                            | 90%         | 38.7 $\pm$ 1.4 | **39.6** $\pm$ 1.6       | **+0.9**   |
> | **EfficientNet-B0** (5.3M) | 50%         | 66.7 $\pm$ 0.6 | 66.4 $\pm$ 1.2           | **-0.3**   |
> |                            | 70%         | 59.2 $\pm$ 1.0 | 59.6 $\pm$ 0.8           | **+0.4**   |
> |                            | 80%         | 55.0 $\pm$ 2.1 | 54.4 $\pm$ 1.4           | **-0.6**   |
> |                            | 90%         | 39.8 $\pm$ 1.9 | 39.7 $\pm$ 3.3           | **-0.1**   |
> | **SwinV2-T** (28.4M)       | 50%         | 51.8 $\pm$ 0.7 | 51.6 $\pm$ 0.9           | **-0.2**   |
> |                            | 70%         | 44.0 $\pm$ 0.8 | 42.6 $\pm$ 1.2           | **-1.4**   |
> |                            | 80%         | 38.1 $\pm$ 0.7 | 39.9 $\pm$ 0.9           | **+1.8**   |
> |                            | 90%         | 27.6 $\pm$ 1.2 | 29.6 $\pm$ 1.5           | **+2.0**   |
>
> **Table 2: Cross-architecture transfer on ImageNet-1K.** Transfer performance using the ResNet-50 embeddings to select coreset for larger target models.
>
> | **Target Model**           | **Pruning** | **Oracle Acc.** | **TopoCore (ResNet-50)** | **Difference** |
> | -------------------------- | ----------- | --------------- | ------------------------ | -------------- |
> | **EfficientNetV2-M** (54M) | 80%         | 39.1 $\pm$ 1.4  | 40.8 $\pm$ 0.3           | **+1.7**       |
> |                            | 90%         | 35.9 $\pm$ 0.3  | 37.1 $\pm$ 1.3           | **+1.2**       |
> | **SwinV2-T** (28M)         | 80%         | 57.8 $\pm$ 0.7  | 59.0 $\pm$ 1.1           | **+1.2**       |
> |                            | 90%         | 38.3 $\pm$ 3.4  | 38.0 $\pm$ 5.1           | **-0.3**       |
> | **SwinV2-B** (88M)         | 80%         | 43.7 $\pm$ 1.2  | 45.8 $\pm$ 2.1           | **+2.1**       |
> |                            | 90%         | 44.4 $\pm$ 2.1  | 45.1 $\pm$ 1.1           | **+0.7**       |
>
> - These results indicate that TopoCore exhibits strong cross-architecture transferability. In many cases (particularly on ImageNet), coresets selected via a standard ResNet proxy actually outperform the Oracle (self-selected) coresets for training advanced architectures like Swin Transformers. This suggests that the topological importance derived from a robust, standard proxy is highly generalizable, allowing users to select a single, high-quality coreset that is effective for training a wide range of target models.
>
>
> > W3: No estimate on the computational cost of the proposed procedure.
>
> -  This is an excellent suggestion. Please refer to Section 2 (Computational complexity and wall-clock analysis) in the Global Response.
>
> > Minor point: Notion of prototype not given until the appendix.
>
> - We appreciate you pointing this out. In the final revision, we will explicitly define the class prototype upon its first appearance in Sections 4.1 as the _barycenter of the point-mass distribution_, ensuring the terminology is clear and self-contained within the main text.

---

### Official Review · Reviewer_FBLQ · 2025-11-14

**Soundness:** 3
**Presentation:** 3
**Contribution:** 3
**Rating:** 6
**Confidence:** 3

**Summary:**

The paper proposes TopoCore, a novel training-free coreset selection framework that leverages topological representations of data rather than purely geometric ones. The method addresses the geometric brittleness of prior approaches that rely on extrinsic distances in feature space.
TopoCore operates in two stages:
1. Global topology-aware manifold embedding using UMAP-like projection to capture intrinsic structure and compute a density-based representativeness score.
2. Local topological optimization via differentiable multi-parameter persistent homology to compute a per-sample persistence score.
The two are combined into a unified TopologyScore for coreset selection. Experiments show that TopoCore outperforms geometric, gradient-based, and score-based baselines, especially under high pruning rates and noisy embeddings. The method also demonstrates strong cross-architecture stability.

**Strengths:**

The paper shows strong originality by unifying manifold learning with differentiable topology for coreset selection, introducing a novel use of persistent homology as an optimization objective. Its technical soundness is reinforced by comprehensive experiments across multiple datasets and architectures, demonstrating consistent gains—especially at high pruning rates. The method is also practically relevant, as it works directly with pretrained embeddings and scales to large real-world settings. Finally, the authors support reproducibility through thorough documentation and a forthcoming anonymous code release.

**Weaknesses:**

1. The paper does not provide a clear runtime or computational complexity analysis of the differentiable multi-parameter persistent homology component, which is generally more expensive than geometric or graph-based baselines. Without wall-clock comparisons or complexity estimates, it is difficult to evaluate the practical feasibility and scalability of TopoCore.
2. The paper reports mean performance but omits confidence intervals or statistical significance tests, making it difficult to assess the claimed stability improvements over baselines.
3. The paper does not clearly state which homology dimensions (e.g., $H_{0}$, $H_{1}$, or higher) are used to compute the persistence-based importance scores. Because different homology groups capture different types of topological structure, this omission makes it harder to interpret what the method is actually measuring.
4. The term “Hilbert decomposition signed measure” is used but is not clearly defined, and it does not appear in the cited reference (Botnan & Lesnick, 2022), making its mathematical meaning and implementation ambiguous. This lack of alignment with standard terminology in multiparameter persistence may confuse readers and hinders reproducibility.
5. Grammar errors:
- Line 171-172: "signifying it's structural importance" -> "signifying its structural importance"
- Line 316-317: "this approach is diverges" -> "this approach diverges"
- Line 995: "Coresest" -> "Coreset"

**Questions:**

1. Could the authors provide runtime measurements or a complexity analysis comparing the computational cost of the differentiable persistent homology component to that of existing coreset baselines, so that the practical scalability of TopoCore can be more clearly understood?
2. Could the authors clarify whether the concept “Hilbert decomposition signed measure” is directly borrowed from Stable Vectorization of Multiparameter Persistent Homology using Signed Barcodes as Measures (Loiseaux et al., 2023), which defines this object formally? In any case, could you include a clear definition for “Hilbert decomposition signed measure”?
3. How sensitive is TopoCore to the choice of UMAP parameters (e.g., Number Neighbors, Minimum Distance)? If you think the choice Number Neighbors=15, Minimum Distance=0.1 is optimal, could you explain?

---

> ### Author Response · Authors · 2025-11-20
>
> Thank you FBLQ for your constructive feedback and positive assessment of our work. We appreciate the clear and actionable suggestions, which have helped us significantly strengthen the manuscript:
>
> ---
> ### Response to Weaknesses/Questions
>
> > W1/Q1: ... runtime measurements or a complexity analysis of the differentiable persistent homology component ... so that the practical scalability of TopoCore can be more clearly understood?
>
> - We agree that assessing practical scalability is crucial. Please refer to Section 2 (Computational complexity and wall-clock analysis) in the Global Response, where we provide a detailed Big-O complexity derivation and empirical runtime benchmarks against geometric baselines.
>
> > W2: Paper reports mean performance but omits confidence intervals ... making difficult to assess claimed stability improvements.
>
> - We appreciate this observation. While we prioritized mean performance in the main text for brevity, standard deviations were reported in Table 5 (Appendix).
> - **Revision:** Our original submission only contained mean values for ImageNet-1K due to compute constraints. Since then, we have completed additional runs to generate valid standard deviation. With the increased page count, we will also prioritize moving these stability metrics into the main Results section (Table 1) in the final revision to better highlight the stability benefits of TopoCore.
>
> > W3: Homology dimensions (e.g., $H_0$, $H_1$, or higher)?
>
> - Thank you for bringing this up. We utilize the $H_1$ homology group (loops) to capture structural features. While this was noted in the hyperparameter table (Appendix Table 8), we agree it must be explicit in the methodology.
> - **Revision:** We will update our notation in Section 3.2 from $\mu_{H(VR_{Y_c}, \hat{f})}^{Hil}$ to $\mu_{H_1(VR_{Y_c}, \hat{f})}^{Hil}$ to explicitly denote dimension 1.
>
> > W4/Q2: "Hilbert decomposition signed measure" not clearly defined ...
>
> - We appreciate you highlighting this. While our original submission included the description: _"This descriptor represents the persistence diagram as a finite collection of positive point masses (representing feature births) and negative point masses (representing feature deaths) in the parameter space of (distance, density),"_ we acknowledge that this text was disconnected from the formal terminology, leading to ambiguity.
> - **Revision:** We will explicitly couple this definition with the term Hilbert decomposition signed measure in Section 3.2 and add the correct citation by Loiseaux et al. [1] to ensure clarity.
>
> > Q3: How sensitive is TopoCore to the choice of UMAP parameters.
>
> - This is an excellent suggestion. We performed a comprehensive sweep of `n_neighbors` (local vs. global structure) and `min_dist` (packing density). Please see Section 3 (Sensitivity to Topology-based hyperparameters) of the Global Response for the detailed results and analysis.
>
> ---
> ### References
>
> [1] D. Loiseaux et al. "Stable vectorization of multiparameter persistent homology using signed barcodes as measures". *NeurIPS*, 2023

---

### Author Response · Authors · 2025-11-12
**Anonymous Code Repository**

We sincerely appreciate all the insightful reviews and are currently preparing detailed responses. In the meantime, in line with our commitment to reproducibility, we have made the anonymized code available [here](https://anonymous.4open.science/r/topocore-anon-C76A/) for your convenience. Individual review responses will follow shortly. Thank you!

---

### Author Response · Authors · 2025-11-20
**Global Response**

We sincerely thank the reviewers for their insightful comments and the opportunity to strengthen our work. In this global response, we address three recurring topics: (1) the suitability of UMAP vs. Topological Autoencoders (Sq9q, y5KN), (2) computational complexity (Sq9q, hXms, FBLQ), and (3) sensitivity to manifold projection hyperparameters (hXms, FBLQ).

> **Revision Plan:** We plan to expand our revision to include dedicated Appendix sections, fully documenting these three analyses. Current manuscript revisions are marked in red.

---

> ### Author Response · Authors · 2025-11-20
> **1. Topological autoencoders as a plugin for UMAP**
>
> We appreciate Sq9q and y5KN’s suggestions to consider Topological Auto-Encoders (TopoAE) [1] or Regularized TDA (RTD) [2] for our *global manifold embedding*. While these methods are powerful for preserving global topology, we chose UMAP for computational efficiency, domain suitability, and alignment with coreset selection goals.
>
> ### A. Computational cost and the "training-free" objective
>
> TopoCore aims to be a lightweight, training-free method applicable to frozen features. UMAP fits this "plug-and-play" requirement perfectly. In contrast, training a topological autoencoder is computationally prohibitive for preprocessing. As shown in Table 1, training RTD on CIFAR-100 takes ~8 hours, whereas UMAP takes ~22 seconds, a speedup of $>1000\times$
>
> **Table 1: Latency(s).** Topological autoencoders require costly training vs. UMAP's algorithmic projection.
>
> |            | Cifar-10 (s) | Cifar-100 (s) |
> | ---------- | ------------ | ------------- |
> | UMAP       | 22.42        | 22.74         |
> | TopoAE [1] | 14,847.79    | 15,248.11     |
> | RTD [2]    | 26,622.89    | 28,661.29     |
>
> ### B. Empirical performance
>
> To empirically validate our choice, we trained both TopoAE and RTD models on CIFAR-10/100 and use their embeddings as a replacement for UMAP in our pipeline. As shown in Table 2, UMAP consistently yields superior or comparable accuracy without the massive training overhead.
>
> **Table 2: Accuracy at 90% Pruning (Avg of 3 runs).** Substituting UMAP with TopoAE [1] or RTD [2] for the global manifold embedding. We utilized  autoencoder training hyperparameters from Table 7 in [2], adopting MNIST settings as RTD provides no CIFAR benchmarks.
>
> | Pruning Rate ($\rightarrow$) | 90%            | 90%            |
> | ---------------------------- | -------------- | -------------- |
> |                              | **Cifar-10**   | **Cifar-100**  |
> | **TopoCore (w/UMAP)**            | 82.1 $\pm$ 0.3 | 45.8 $\pm$ 0.7 |
> | **TopoCore (w/TopoAE [1])**      | 75.0 $\pm$ 0.3 | 41.2 $\pm$ 1.0 |
> | **TopoCore (w/RTD [2])**         | 78.0 $\pm$ 1.7 | 46.4 $\pm$ 0.4 |
>
> ### C. Why this accuracy drop? Domain suitability and structural alignment
>
> The observed drop in coreset accuracy in Table 2, when using topological autoencoders, likely stems from a fundamental misalignment between their objectives:
>
> - **Latent Space Quality:** TopoAEs struggle to produce clean, separated latent representations for complex, colored datasets like CIFAR-10/100. As noted in the TopoAE paper itself, CIFAR-10 is _"challenging to embed... in a purely unsupervised manner"_ (Section 5.2.2 in [1]), often resulting in latent spaces where classes are homogeneously mixed rather than cleanly separated (see Figure 4 in [1]). This lack of separation severely hampers the effectiveness of our per-class density estimation, unlike the distinct cluster delineation achieved by UMAP.
> - **Global vs. Local Fidelity:** This issue is exacerbated by the fact that TopoAE and RTD prioritize preserving _global_ structural similarity (e.g., maintaining relative distances between distinct mammoth "head" and "foot" clusters as shown in Figure 1 in [2]). While this global constraint is valuable for visualization, it is less relevant for coreset selection, where we partition the data into class-based manifolds.
>
> > In summary, while topological autoencoders are robust tools for manifold learning, UMAP provides a more efficient, domain-appropriate, and higher-performing foundation for our specific dual-scale framework. We encourage future work exploring topological autoencoders that flexibly balance global and local structural priorities, specifically optimized for the task of point-cloud sparsification.
>
> *Compute for this analysis:* Autoencoder training, runtime and accuracy were performed on 1x NVIDIA A40 GPU and 1x AMD EPYC 7502 32-Core CPU. Runtime was calculated with no background tasks running.
>
> ---
> ### References
>
> [1] M. Moor et al. "Topological autoencoders". *ICLR*, 2020. \
> [2] I. Trofimov et al. "Learning topology-preserving data representations". *ICLR*, 2023.

---

> ### Author Response · Authors · 2025-11-20
> **2. Computational complexity and wall-clock analysis**
>
> ### A. Complexity analysis
>
> We thank Sq9q, hXms, and FBLQ for this request. Below, we detail the theoretical complexity of TopoCore relative to geometric baselines, following analyses from [3].
>
> **Table 3: Computational complexity.** Comparing complexity of geometry-based methods.
>
> |          | Computational Complexity                                          | Explanation                                           |
> | -------- | ----------------------------------------------------------------- | ----------------------------------------------------- |
> | Moderate | $\mathcal{O}(Nd + N_c \log N_c)$                                  | Distance calc. ($Nd$) + Prototype sorting             |
> | D2       | $\mathcal{O}(Nkd) + \mathcal{O}(HNk)$                             | kNN graph ($Nkd$) + Message passing ($H$ steps)       |
> | TopoCore | $\mathcal{O}(N \log N) + C * H * [m * \mathcal{O}(N_c \log N_c)]$ | Global UMAP + Local persistence ($H$ steps, grid $m$) |
>
>  $N$ is the dataset size with $C$ classes, $N_c \approx N/C$ samples per class, $d$ dimension, and $k$ neighbors. For TopoCore, cost is dominated by the grid resolution $m$ used for the Kernel Density Estimator.
>
> - Crucially, we utilize the Hilbert decomposition signed measure, which reduces the multi-parameter problem to *one-parameter persistence slices along a grid*. As detailed in Appendix D.1 of [4], for a 2-parameter filtration (Rips + Density) on a grid of size $m$, the algorithm performs $m$ *runs of a 1-parameter persistence optimization*.
> - Therefore, the persistence homology optimization is the cost of a 1-parameter optimization on $N_c$ points. While the theoretical worst-case for persistence is cubic [5], in the 1-parameter persistence case the computation is empirically linear  $\mathcal{O}(N_c \log N_c)$ [6].
> - Furthermore computing $\mathcal{L}_{\text{pers}}(Y_c)$
> (Equation 3 in our paper), simplifies to summing the feature persistences (see Corollary E.2 [7]) which is bounded by a constant $B$ derived from the simplicial complex $K$, making the backward pass $\mathcal{O}(1)$. Thus, the local optimization cost is strictly $m \cdot \mathcal{O}(N_c \log N_c)$.
>
> > In summary, our method remains tractable because we use a construction which reduces the 2-parameter persistence problem to a sequence of $m$ standard 1-parameter persistence calculations [4]. By also applying this per-class (where $N_c$ is small) and utilizing the efficient reduction strategy, we avoid prohibitive costs typically associated with multiparameter topology.
>
> ### B. Empirical wall-clock analysis
>
> We also benchmark latency and resource utilization on an AMD EPYC 7502 (32-Core) CPU.
>
> **Table 4: Latency and Utilization.** Performing selection on CIFAR-100.
>
> | **Method**   | **Global Step (s)** | **Local Step (s)** | **Total (s)** | **Max CPU Util.** |
> | ------------ | ------------------- | ------------------ | ------------- | ----------------- |
> | **TopoCore** | 22.74               | 749.08             | **771.82**    | **16%**           |
> | **D2**       | -                   | -                  | **86.05**     | **82%**           |
>
> Despite higher latency, TopoCore under-utilizes hardware (16% vs. 82%), indicating an implementation-specific bottleneck. Our backend (`multipers`) currently lacks multi-processing/GPU support. We anticipate that future parallelization efforts will ultimately establish topology-based descriptors as a compelling and practical tool. Regardless, TopoCore offers superior a cost-benefit:
>
> - It is significantly faster than training-dynamic coreset methods, which require training a proxy model from scratch (hours of compute) versus our probe of frozen embeddings (minutes).
> - Unlike geometric heuristics (D2/Moderate) which suffer from high variance or lower accuracy, TopoCore accepts a higher upfront cost to guarantee a stable, high-fidelity coreset, eliminating the need for repeated selection runs.
>
> > In summary, we acknowledge the current performance gaps inherent in utilizing differential topology. We view this work as a foundational step in elucidating potential applications and applying differential persistence to probe the embedding space of neural networks.As persistence libraries mature to leverage parallel architectures, the wall-clock gap will close, offering the superior stability of topological methods with negligible latency trade-offs.
>
> ---
> ### References
>
> [3] Nagaraj et al. "Coresets from trajectories: Selecting data via correlation of loss differences." *ArXiv*, 2025 \
> [4] Loiseaux et al. "Stable vectorization of multiparameter persistent homology using signed barcodes as measures". *NeurIPS*, 2023 \
> [5] Lesnick et al. "Computing minimal presentations and bigraded Betti numbers of 2- parameter persistent homology". *SIAGA*, 2022 \
> [6] Bauer et al. "Keeping it sparse: Computing persistent homology revisited." _ArXiv_, 2022 \
> [7] Scoccola et al. "Differentiability and optimization of multiparameter persistent homology". *ICML*, 2024

---

> ### Author Response · Authors · 2025-11-20
> **3. Sensitivity to Topology-based hyperparameters**
>
> We thank hXms and FBLQ for the valuable suggestion to perform a UMAP hyperparameter sweep. This study is crucial for establishing the robustness of TopoCore. Below we provide a sweep of two hyperparameters: (1) `n_neighbors` which controls the trade-off between global and local geometric preservation and (2) `min_dist` which controls how tightly samples are packed in the low-dimensional manifold. We trained *5 models* per hyperparameter combination using a 90% pruning rate on CIFAR-100. Table 1 reports the average accuracy ($\pm$ standard deviation), and Table 2 shows the performance change relative to our default configuration.
>
> **Table 5: TopoCore Accuracy on CIFAR-100 (90% Pruning).** Columns represent `n_neighbors` and row represent `min_dist`.
>
> |                     | 5              | ***15***             | 50             | 100            | Average over Rows |
> | ------------------- | -------------- | -------------------- | -------------- | -------------- | ----------------- |
> | 0.05                | 43.9 $\pm$ 2.2 | 43.0 $\pm$ 0.9       | 44.4 $\pm$ 1.7 | 44.9 $\pm$ 1.6 | 44.1 $\pm$ 1.6    |
> | ***0.1***           | 43.1 $\pm$ 1.8 | ***45.8 $\pm$ 0.7*** | 45.7 $\pm$ 0.8 | 43.6 $\pm$ 0.7 | 44.6 $\pm$ 0.9    |
> | 0.5                 | 46.2 $\pm$ 1.0 | 43.8 $\pm$ 0.7       | 44.3 $\pm$ 2.2 | 43.9 $\pm$ 0.4 | 44.6 $\pm$ 1.0    |
> | 1                   | 44.3 $\pm$ 0.9 | 45.4 $\pm$ 0.6       | 45.0 $\pm$ 1.1 | 43.1 $\pm$ 1.3 | 44.4 $\pm$ 1.0    |
> | Average over Column | 44.4 $\pm$ 1.5 | 44.5 $\pm$ 0.7       | 44.8 $\pm$ 1.4 | 43.9 $\pm$ 1.0 |                   |
>
> **Table 6: Accuracy change** relative to the default configuration of `n_neighbors=15` and `min_dist=0.1`.
>
> |      | 5    | 15         | 50   | 100  |
> | ---- | ---- | ---------- | ---- | ---- |
> | 0.05 | -1.9 | -2.8       | -1.4 | -0.9 |
> | 0.1  | -2.7 | $\bigstar$ | -0.1 | -2.2 |
> | 0.5  | +0.4 | -2.0       | -1.5 | -1.9 |
> | 1.0  | -1.5 | -0.4       | -0.8 | -2.7 |
>
> We see that our selection of `n_neighbors=15` and `min_dist=0.1` provides a strong balance of high accuracy and low variance (baring the one outlier of `n_neighbors=5` and `min_dist=0.5`). It's also important to note that the column for `n_neighbors=15` exhibits the lowest average standard deviation (0.7) across all `min_dist` settings. This indicates that our choice of 15 neighbors provides the most stable coreset quality.
>
> > In summary, this additional study closes the loop on the sensitivity analysis of TopoCore. In our original submission, we provided ablations on the persistence optimization steps (Appendix A.4) and the $\alpha / \beta$ scoring terms (Appendix A.5). With this final ablation on UMAP parameters, we have now demonstrated robustness across all three components of our method: (a) Global manifold embedding, (b) Local persistence, and (c) the combined importance score.

---

### Meta-Review · Area_Chair_Vzu5 · 2026-01-07

**Summary:**

Here is a summary of the reviewers' concerns.

Compute and  scalability still a central risk (FBLQ, Sq9q, hXms).
 - Principal complaint: no credible cost–benefit picture without runtime/memory scaling vs baselines.

Method clarity / definition debt is only partially repaid (FBLQ, y5KN, hXms).
 - Ambiguous or missing definitions and algorithmic flow (notably “Hilbert decomposition signed measure”, objective direction, and the full end-to-end selection procedure) (FBLQ, y5KN).

Topology score behavior, prototypes vs outliers remains conceptually delicate (y5KN)
 - Persistent homology can assign high persistence to structurally isolated points, y5KN explicitly worries high persistence for outliers conflicts with coreset intuition (“pick prototypical points”).

 Statistical evidence remains somewhat weak (FBLQ, y5KN, hXms)
 - Means without CIs/tests; “slightly better” improvements often not significant, claims like “4\times better precision” need consistent tabular backing (FBLQ, y5KN, hXms).


Generality and  projector dependence remain open (Sq9q, hXms, FBLQ)
 - UMAP choice, reviewers ask why UMAP, sensitivity to $ (n_{neighbors}, \text{min}_{dist}) $, and comparisons to topology-preserving representation methods (Sq9q, FBLQ, hXms).
 - Modality breadth. Evaluation largely vision-only, hXms wanted NLP (IMDB/ANLI) and kept this as a limitation even after rebuttal (hXms).

**Reviewer Concerns:**

Reviewer concerns which  are still outstanding after rebuttal.

Compute and  scalability  (FBLQ, Sq9q, hXms)
 - After rebuttal, even with added wall-clock, reviewer hXms concludes TopoCore is ~order-of-magnitude slower than D2 on CIFAR-100, so practicality remains questionable despite accuracy/stability gains (hXms).
 - Need scaling curves (e.g., N, C, embedding dim), kNN + PH breakdown, and evidence the slowdown is not fundamental (or a concrete acceleration plan).

Method clarity and definition debt is only partially repaid (FBLQ, y5KN, hXms)
 - After rebuttal authors promise clearer pipeline + corrected citations/notation, but y5KN still flags presentation as a major obstacle, requesting a formal algorithm and resolving “minimize vs increase persistence” contradictions (y5KN).
 - Readers still may not be able to reconstruct the method unambiguously, a clean pseudocode + explicit objective statement is still needed.

Topology score behavior: prototypes vs outliers remains conceptually delicate (y5KN)
 - Still needs quantitative evidence that the combined score selects prototypes rather than boundary noise across datasets, and that the density term reliably suppresses outliers without erasing useful boundary points.

Statistical evidence remains somewhat weak (FBLQ, y5KN, hXms)
 - Still missing formal significance testing (or at least paired tests / bootstrap CIs) and a consistent reporting standard across all main claims.

Generality and projector dependence remain open (Sq9q, hXms, FBLQ)
 -  hXms wanted NLP (IMDB/ANLI) and kept this as a limitation even after rebuttal (hXms).
 - Need either (i) convincing projector-agnostic results (UMAP vs alternatives) or (ii) clear guidance + robustness bounds, and at least one non-vision benchmark to support “task-agnostic” claims.

One-line remaining blockers by reviewer
 - (FBLQ): compute analysis + statistical uncertainty + missing technical specification/definitions.
 - (Sq9q): compare to newer topology-preserving embeddings, broaden architectures/tasks, quantify cost.
 - (y5KN): clarity/algorithm, objective direction inconsistencies, persistence/outlier behavior, significance/practicality.
 - (hXms): still slow in practice, still vision-only, some claims/sensitivities not fully nailed down.

**Reviewer Scores:**

Most probably the scores would have remained unchanged

---

### Decision · Program_Chairs · 2026-01-26

Reject